# From alternative conceptions of honesty to alternative facts in communications by US politicians

Jana Lasser [1,2], Segun T. Aroyehun [1,3], Fabio Carrella [4], Almog Simchon [4], David Garcia [1,2,3] & Stephan Lewandowsky [4,5,6] ✉

The spread of online misinformation on social media is increasingly perceived as a problem for societal cohesion and democracy. The role of political leaders in this process has attracted less research attention, even though politicians who 'speak their mind' are perceived by segments of the public as authentic and honest even if their statements are unsupported by evidence. By analysing communications by members of the US Congress on Twitter between 2011 and 2022, we show that politicians' conception of honesty has undergone a distinct shift, with authentic belief speaking that may be decoupled from evidence becoming more prominent and more differentiated from explicitly evidence-based fact speaking. We show that for Republicans—but not Democrats—an increase in belief speaking of 10% is associated with a decrease of 12.8 points of quality (NewsGuard scoring system) in the sources shared in a tweet. In contrast, an increase in fact-speaking language is associated with an increase in quality of sources for both parties. Our study is observational and cannot support causal inferences. However, our results are consistent with the hypothesis that the current dissemination of misinformation in political discourse is linked to an alternative understanding of truth and honesty that emphasizes invocation of subjective belief at the expense of reliance on evidence.

Numerous indicators suggest that democracy is in retreat worldwide (for example, refs. 1,2). Although symptoms and causes of this democratic backsliding are difficult to tease apart, the widespread dissemination of misinformation—on social media, in hyper-partisan news sites and in political discourse—is undoubtedly a challenge to democracies[3]. There is increasing evidence that exposure to misinformation can cause people to change their behaviour (for example, ref. 4). Exposure to misinformation has been identified as a contributing cause of voting for populist parties in Italy[5] and has been causally linked to ethnic hate crimes in Germany[6] (for a review of causal effects, see ref. 7). Note that we use 'misinformation' as an umbrella term to refer to any information

that people consume that later turns out to be false. Misinformation can be spread unintentionally, when communicators mistakenly believe some item of information to be true, or it can be spread intentionally, for example, in pursuit of a political agenda. Intentionally disseminated misinformation is often referred to as 'disinformation'. The psychological and cognitive consequences of disinformation are indistinguishable from those of unintentional misinformation, and we therefore use the latter term throughout.

Misinformation has several troubling psychological attributes. First, misinformation lingers in memory even if people acknowledge, believe and try to adhere to a correction[8]. Although people may adjust

[1]Graz University of Technology, Graz, Austria. [2]Complexity Science Hub Vienna, Vienna, Austria. [3]University of Konstanz, Konstanz, Germany. [4]University of Bristol, Bristol, UK. [5]University of Western Australia, Crawley, Western Australia, Australia. [6]University of Potsdam, Potsdam, Germany. ✉e-mail: stephan.lewandowsky@bristol.ac.uk

their factual beliefs in response to corrections[9], their political behaviours and attitudes may be largely unaffected[10,11]. Second, perhaps most concerningly, in some circumstances people may even come to value overt dishonesty as a signal of 'authenticity'[12]. A politician who routinely and blatantly misinforms the public is overtly violating the established societal norm of being accurate and truthful. This norm violation can identify the politician as an enemy of the 'establishment' and, by implication, an authentic champion of 'the people'—dishonesty and misinformation thus become a sign of distinction[12]. For example, polls have shown that around 75% of Republicans considered President Trump to be 'honest' at various points throughout his presidency (for example, NBC poll, April 2018). This perception of honesty is at odds with the records of fact checkers and the media, which have identified more than 30,000 false or misleading statements by Trump during his presidency (*Washington Post* fact checker[13]).

This discrepancy between factual accuracy and perceived honesty is, however, understandable if 'speaking one's mind' on behalf of a constituency is considered a better marker of honesty than veracity. The idea that untrue statements can be honest, provided they arise from authentic belief speaking, points to a distinct ontology of honesty that does not rely on the notion of evidence but on a radically constructivist appeal to an intuitive shared experience as 'truth'[3]. There have been several attempts to characterize this ontology of truth and honesty and the stream of misinformation to which it gives rise (for example, refs. 3,14,15). A recent analysis[3] of ontologies of political truth (see also ref. 16) proposed two distinct conceptions of truth: 'belief speaking' and 'fact speaking'. Belief speaking relates only to the speaker's beliefs, thoughts and feelings, without regard to factual accuracy. Fact speaking, in contrast, relates to the search for accurate information and an updating of one's beliefs based on that information.

The first of these two ontologies echoes the radical constructivist truth, based on intuition and feelings, that also characterized 1930s fascism (for example, ref. 17). This conception of truth sometimes rejects the role of evidence outright. For example, Nazi ideology postulated the existence of an 'organic truth' based on personal experience and intuition that can only be revealed through inner reflection but not external evidence (for example, refs. 17,18). Contemporary variants of this conception of truth can be found in critical postmodern theory[19] and both right-wing and left-wing populism[20,21]. The second ontology, based on fact speaking, aims to establish a shared evidence-based reality that is essential for the well-being of democracy[22]. This conception of truth aims to be dispassionate and does not admit appeals to emotion as a valid tool to adjudicate evidence, although it also does not preclude truth-finding from being highly contested and messy (ref. 23 versus ref. 24).

For democratic societies, a conception of truth that is based on belief speaking alone can have painful consequences because democracy requires a body of common political knowledge to enable societal coordination[22]. For example, people in a democracy must share the knowledge that the electoral system is fair and that a defeat in one election does not prevent future wins. Without that common knowledge, democracy is at risk. The attempts by Donald Trump and his supporters to overturn the 2020 election results with baseless claims of electoral fraud have brought that risk into sharp focus[25]. To achieve a common body of knowledge, democratic discourse must go beyond belief speaking. In particular, democratic politics requires fact speaking by leaders—otherwise, they may choose to remain wilfully ignorant of embarrassing information, for example, by refusing briefings from experts that are critical of their favoured public-health policy. A corollary of this requirement is that the public considers fact speaking by politicians as an indicator of honesty rather than (only) belief speaking.

Although truth and honesty are closely linked concepts, with honesty and truthfulness being nearly synonymous[26], in the present context they need to be disentangled for clarity. Here we focus primarily on conceptions of honesty, which refers to a virtuous human quality

and a socially recognized norm, rather than truth, which refers to the quality of information about the world. Thus, the two ontologies of truth just introduced describe how the world can be known—namely, either through applying intuition or seeking evidence, irrespective of the virtuous qualities (or lack thereof) of the beholder. This ontological dichotomy maps nearly seamlessly into the different conceptions of honesty that we characterize as belief speaking and fact speaking.

So far, there has been much concern but limited evidence about the increasing prevalence of belief speaking at the expense of fact speaking in American public and political life. We aim to explore this presumed shift in conceptions of truth and honesty by focusing on Twitter activity by members of both houses of the US Congress. The United States is not only one of the world's largest democracies but it is also a crucible of the contemporary conflict between populism and liberal democracy and the intense partisan polarization this conflict has entailed[27]. The choice of Twitter is driven by the fact that public outreach on Twitter has become one of the most important avenues of public-facing discourse by US politicians in the last decade[28] and is frequently used by politicians for agenda-setting purposes[29].

Our analysis addressed several research questions: can we identify aspects of belief speaking and fact speaking in public-facing statements by members of Congress? If so, how do these conceptions evolve over time? What partisan differences, if any, are there? Is the quality of shared information linked to the different conceptions of honesty? To answer these questions, we performed a computational analysis of an exhaustive dataset of tweets posted by US politicians, detecting links to misinformation sources and analysing text from tweets and news sources.

## Identifying conceptions of honesty in political speech

We first sought to identify the two components of truth and honesty—belief speaking and fact speaking—in public-facing political speech by elected US officials. For our analyses, we collected a corpus of tweets from members of the US Congress between 1 January 2011 and 31 December 2022. After removing retweets and duplicates, our corpus contained a total of 4,527,814 tweets (see Methods for details). Twitter accounts were categorized by party affiliation.

To measure the conceptions of honesty in text, we created two dictionaries of words associated with each of the concepts. We followed a computational grounded theory approach[30] to incorporate both expert knowledge and computational pattern recognition. We started with a list of seed words for each conception, followed by computational expansion and iterative pruning and refinement through human input (see Methods for details).

We validated the dictionaries in three steps. First, to validate the candidate keywords (selected by the authors), we created a survey on Prolific and asked participants ($N = 51$) to rate each keyword's representativeness of the two honesty components on two separate Likert scales. We then ran paired *t*-tests between each word's representativeness ratings for belief speaking and fact speaking, respectively. Keywords that were rated as significantly more representative for belief speaking (fact speaking) were included in the belief-speaking (fact speaking) dictionaries. The final dictionaries include a total of 37 keywords for each component and are provided in Extended Data Table 1 (see Methods and Supplementary Notes 1 and 2 for details). Following the distributed dictionary representation approach[31], we converted the keywords into vector embeddings using a pre-trained algorithm (GloVe). These representations capture nuanced contextual information and are amenable to a vector-similarity approach to establish overlap between each dictionary and the text or document of interest (see Methods for details).

In the second validation step, we applied the dictionaries to our tweet corpus and calculated the semantic similarity $D_b$ and $D_f$ between the article and the belief-speaking and fact-speaking dictionaries,

respectively (see Methods for details). A positive semantic similarity means that a piece of text is more similar to the words contained in a dictionary, whereas a negative similarity means that it is more dissimilar. We then sampled tweets that had a high belief-speaking or fact-speaking similarity or were dissimilar to both honesty components. We again created a survey on Prolific with the same set-up as described for the keyword validation. Using tweets that a majority of human raters agreed were representative of belief speaking or fact speaking as ground truth, we found satisfactory agreement between the computed belief-speaking and fact-speaking similarity scores and human ratings, with AUC = 0.824 for belief speaking and AUC = 0.772 for fact speaking (see Methods and Supplementary Note 3 for details).

In the third validation step, we applied the dictionaries to historical articles from the *New York Times* (*NYT*) for three text categories: 'opinion', 'politics' and 'science' (see Methods for details). We found that articles in the science category are more similar to fact speaking than all articles on average ($< D_f >_{sci} - < D_f >= 0.033$), followed by articles in the opinion ($< D_f >_{op} - < D_f >= 0.006$) and politics ($< D_f >_{pol} - < D_f >= -0.006$) categories. Articles in the opinion category show the highest similarity to the belief-speaking dictionary ($< D_b >_{op} - < D_b >= 0.013$), followed by articles in the science ($< D_b >_{sci} - < D_b >= 0.009$) and politics ($< D_b >_{pol} - < D_b >= -0.007$) categories. The analysis of *NYT* content confirmed our expectation of articles in the science category being most similar to fact speaking, whereas articles in the opinion category being most similar to belief speaking. It did not confirm our expectation of politics being more similar to fact speaking than opinion articles and more similar to belief speaking than science articles.

Finally, to establish the uniqueness of our dictionaries and to differentiate the honesty conceptions from existing similar measures, we investigated the relationship between our two components to text features such as authenticity[32], analytic language[33] and a moral component reflecting judgemental language[34], each measured using Linguistic Inquiry and Word Count 2022 (LIWC-22) (ref. [35]), and positive and negative sentiment measured using VADER[36]. We calculated scores for each of these components for every tweet in the corpus. Both belief speaking and fact speaking are negatively correlated with 'analytic', although the correlation with belief speaking ($r = -0.27$) is about twice as high as with fact speaking ($r = -0.16$). Both honesty components are positively correlated with 'authentic', 'moral' and negative sentiment, whereas the correlation with positive sentiment is positive for belief speaking ($r = 0.06$) and sightly negative for fact speaking ($r = -0.01$). All correlations are highly significant ($P < 0.001$) but small—the correlation with the largest magnitude ($r = -0.27$) is observed between belief-speaking similarity and 'analytic'. Details of the comparison with LIWC and VADER scores are summarized in Supplementary Note 4. In summary, these analyses show that belief speaking and fact speaking do not overlap greatly with existing related measures of text features.

## Partisan and temporal dynamics of conceptions of honesty

After validating our dictionaries, we produced textual scatterplots[37] (see Methods for details) to illustrate individual terms that are characteristic of the two honesty components.

Figure 1 shows diagnostic words in a two-dimensional plot, with the *x* and *y* axes representing party and honesty conception, respectively. Each dot is a unigram from the Twitter corpus, and its colour is associated with party keyness (a word with positive party keyness occurs more often for texts from members of a given party than expected by chance). The closer to a corner a word is, the more it characterizes that particular conception of honesty and party dimension. See Methods for details on how words in the figure are represented. We see that Republican belief-speaking keywords, situated in the top-left corner, often refer to political opponents or ideologies ('Biden', 'democrats' and 'conservatives') or conservative values ('freedom' and 'liberty').

In contrast, fact-speaking keywords by the same party are linked to economic ('energy', 'taxpayer' and 'trade') or foreign-policy aspects ('China' and 'Chinese') and the military. On the right-hand side of the figure, we find that Democrat belief-speaking tweets also regard politicians and political ideology ('Trump', 'Democrats' and 'Republicans') and social justice ('colour', 'discrimination' and 'justice'), whereas fact-speaking texts particularly concern the climate crisis ('climate') and social welfare and healthcare ('worker', 'care' and 'pre-existing condition').

Supplementary Note 6 explores the topics of politicians' communications further. Analysis of some controversial topics showed that these topics invoked more belief speaking or fact speaking than the average tweet, with only a few exceptions. For example, vaccine-related discourse involved far less belief speaking than other controversially discussed topics such as climate change or the opioid crisis for both parties.

We next examined the temporal trends of the two honesty components. For the following analyses, we use the centred and length-corrected belief-speaking and fact-speaking similarity scores $D'_b$ and $D'_f$ (see Methods for details). To arrive at a finer-grained picture of the variability of these components between individual politicians, we calculated the average belief-speaking similarity $< D'_b >_{acc}$ and fact-speaking similarity $< D'_f >_{acc}$ of tweets for each individual politician. Note that $<>_{acc}$ denotes an average over all posts from a given Twitter account. Figure 2a–d shows how the distribution of $< D'_b >_{acc}$ and $< D'_f >_{acc}$ shifted between the first (2011–2013, 331 Democrats and 514 Republicans) and last (2019–2022, 295 Democrats and 494 Republicans) 4 years of tweets contained in the corpus.

For both parties, the mean belief-speaking similarity $< D'_b >_{party}$ increased considerably from −0.031 to 0.017 for Democrats (unpaired *t*-test $t(558) = -11.317$, $P < 0.001$, Cohen's $d = 0.850$, 95% confidence interval (CI) of difference in means [−0.06, −0.04]) and from −0.040 to 0.012 for Republicans ($t(493) = -10.819$, $P < 0.001$, $d = 0.854$, 95% CI = [−0.06, −0.04]). Similarly, we see an increase in the similarity to truth-seeking $< D'_t >_{party}$ from −0.027 to 0.009 for Democrats ($t(516) = -9.753$, $P < 0.001$, $d = 0.748$, 95% CI = [−0.04, −0.03]) and from −0.038 to −0.003 for Republicans ($t(483) = -8.442$, $P < 0.001$, $d = 0.671$, 95% CI = [−0.04, −0.03]). This overall increase in both belief-speaking and truth-seeking similarity also becomes apparent in Fig. 2e,f, and is especially pronounced after the presidential election in late 2016.

This parallel increase for both belief speaking and fact speaking could reflect the fact that, in recent years, topics concerning fake news have become increasingly central to political discourse[38], resulting in opposing claims and counterclaims (for example, Donald Trump routinely accused mainstream media such as the *NYT* of spreading 'fake news'[29]). Whereas those claims mainly represented belief speaking, they were accompanied by increasing attempts by the media, and other actors, to correct misinformation through fact-speaking discourse.

## Relating honesty components to information trustworthiness

To test our hypothesis that belief speaking is preferentially associated with dissemination of misinformation, we analysed the association between belief speaking and fact speaking to the quality of the information that is being relayed. To assess information quality, we examined links to websites external to Twitter that were shared by the accounts. We followed an approach used in similar research in this domain[39,40] and used a trustworthiness assessment by professional fact checkers of the domain to which a link points. We used the NewsGuard information nutrition database[41] and an independently compiled database of domain trustworthiness labels[42] (see Methods and Supplementary Notes 7 and 8 for details).

As of the beginning of March 2022, the NewsGuard database indexed 6,860 English language domains. Each domain is scored on a total of 9 criteria, ranging from 'does not label advertising' to 'repeatedly publishes false information'. Each category awards a varying

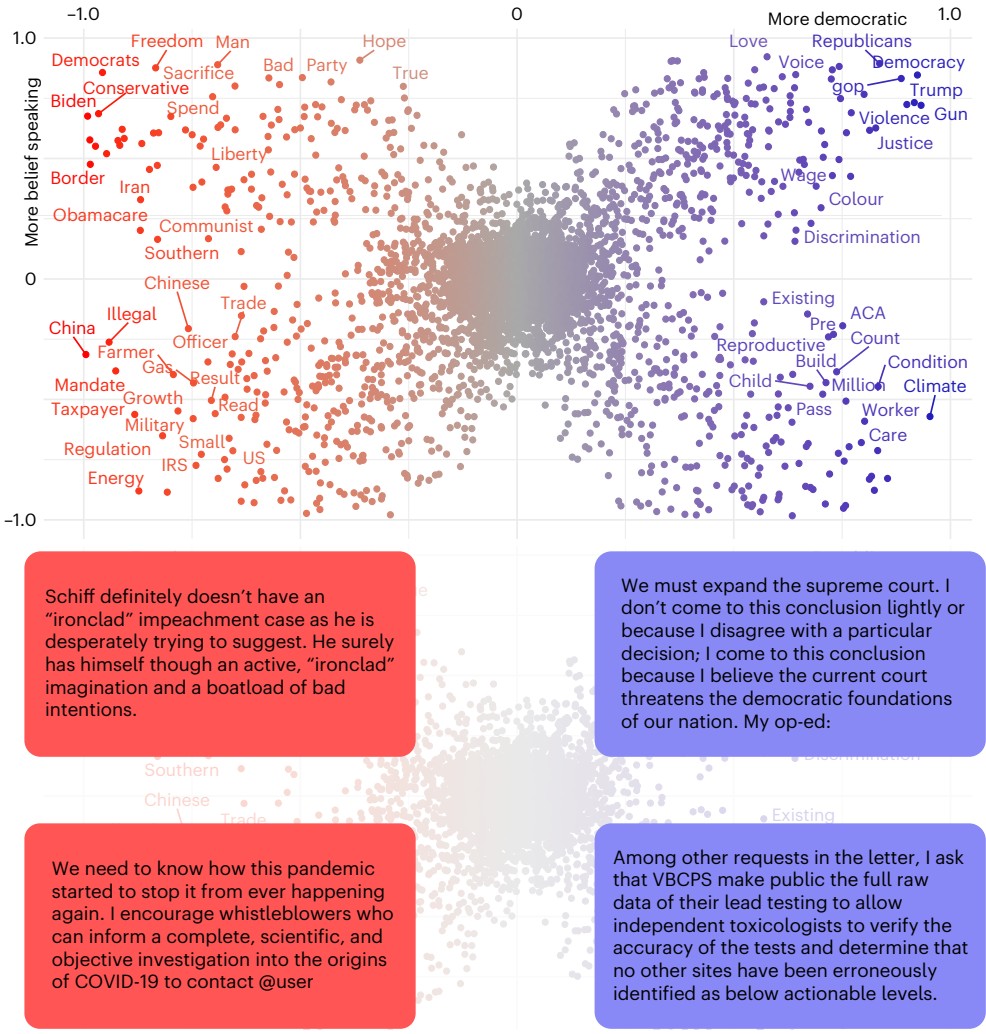

**Fig. 1 | The figure depicts the distribution of keywords on a textual scatterplot.** Each term is a dot with two coordinates associated with party (x coordinate) and honesty-component (y coordinate) keyness. Each coordinate represents an SFS value ranging from −1 to 1. The colour of the word is associated with party keyness. We only show word labels where SFS > 0.65 or SFS < −0.65

for readability reasons. Below the scatterplot we show four example tweets associated with the four quadrants of the scatterplot. ACA, Affordable Care Act; COVID-19, coronavirus disease 2019; IRS, Internal Revenue Service; VBCPS, Virginia Beach City Public Schools.

number of points for a total of 100. Domains with less than 60 points are considered 'not trustworthy'. The majority of indexed domains (63%) are considered trustworthy. After excluding links to other social media platforms (for example, Twitter, Facebook, YouTube.com and Instagram) and links to search services (Google and Yahoo), the database covered between 20% and 60% of the links posted by members of the US Congress, with a steadily increasing share of links covered over time and no difference in coverage between the parties (Extended Data Fig. 1).

For each tweet, we calculated the belief-speaking and fact-speaking similarity $D'_b$ and $D'_f$. Figure 3a,b shows the NewsGuard score, $S'_{NG}$, rescaled to [0; 1] over the belief-speaking and fact-speaking similarity, respectively, for each tweet posted by a member of Congress. Note that in Fig. 3 the y axis appears truncated to improve legibility. The full data are shown in Extended Data Fig. 2.

To investigate the relationship between $D'_b$, $D'_f$ and $S'_{NG}$, we fitted a linear mixed-effects model with random slopes and intercepts for every Congress member following equation (1). The lines shown in Fig. 3a,b show $S'_{NG}$ predicted by the model depending on $D'_b$, $D'_f$, respectively, party $P$ and their interaction terms (see Methods for details).

The analysis conducted with $P$ = Democrat as baseline yielded a significant fixed effect for $D'_f$ ($t(504, 809) = 3.6$, $P < 0.001$, coefficient

0.022, 95% CI = [0.010, 0.033]), $P$ = Republican ($t(504, 809) = −29.9$, $P < 0.001$, coefficient −0.069, 95% CI = [−0.074, −0.065]), the interaction between Republican and $D'_b$ ($t(504, 809) = −14.4$, $P < 0.001$, coefficient −0.128, 95% CI = [−0.146, −0.111]), the interaction between Republican and $D'_f$ ($t(504, 809) = 9.6$, $P < 0.001$, coefficient 0.085, 95% CI = [0.068, 0.103]), and the three-way interaction between $D'_b$, $D'_f$ and Republican ($t(504, 809) = −5.6$, $P < 0.001$, coefficient −0.085, 95% CI = [−0.115, −0.056]). See Extended Data Table 2 for the full regression statistics and Extended Data Fig. 3 for a visualization of the fixed effect of the three-way interaction.

Therefore, an increase in $D'_b$ of 10% predicted a decrease in News-Guard score of 12.8, but only for members of the Republican party. An increase in $D'_f$ of 10% predicted an increase in NewsGuard score of 2.1 for Democrats and of 10.6 for Republicans. For Democrats, we find no significant relationship between $S_{NG}$ and belief-speaking similarity. Predictions of the NewsGuard score depending on belief-speaking and fact-speaking similarity based on the two-way interactions between honesty components and party are shown as lines in Fig. 3a and Fig. 3b, respectively

In Supplementary Note 9, we explore this pattern further by considering NewsGuard scores and honesty components broken down by

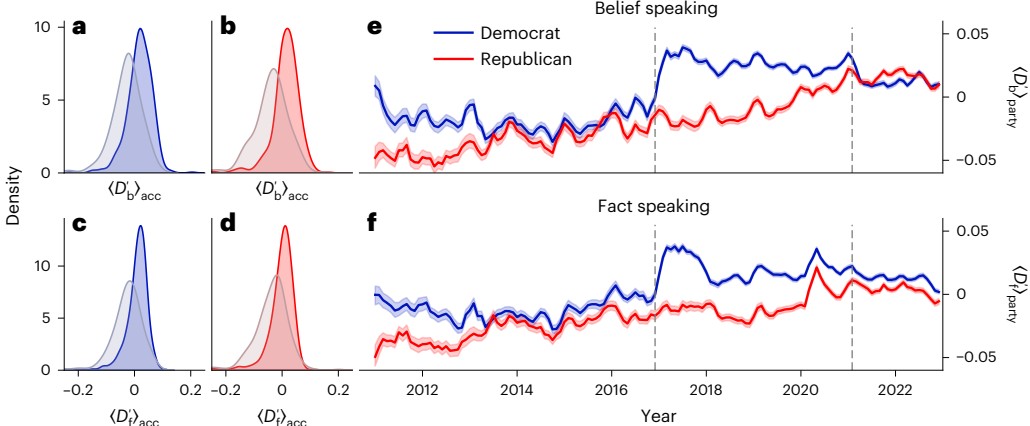

**Fig. 2 | Belief-speaking and fact-speaking similarity in tweets by members of the US Congress for the period 2011–2022 shown separately for members of each party. a,b**, Distributions of the average within-politician belief-speaking similarity $< D'_b >_{acc}$ in tweets of members of the Democratic (**a**) and Republican (**b**) parties for the years 2011–2013 (grey) and 2019–2022. **c,d**, Distributions of the average within-politician fact-speaking similarity $< D'_f >_{acc}$ for Democratic (**c**) and Republican (**d**) members. **e,f**, Shown are the micro averages over all tweets of belief-speaking similarity $< D'_b >_{party}$ (**e**) and fact-speaking similarity $< D'_f >_{party}$ (**f**) over time. Timelines have been smoothed with a rolling average of 3 months. The 95% CIs were computed with bootstrap sampling over 1,000 iterations. Dashed vertical lines indicate dates of presidential elections in 2016 and 2020.

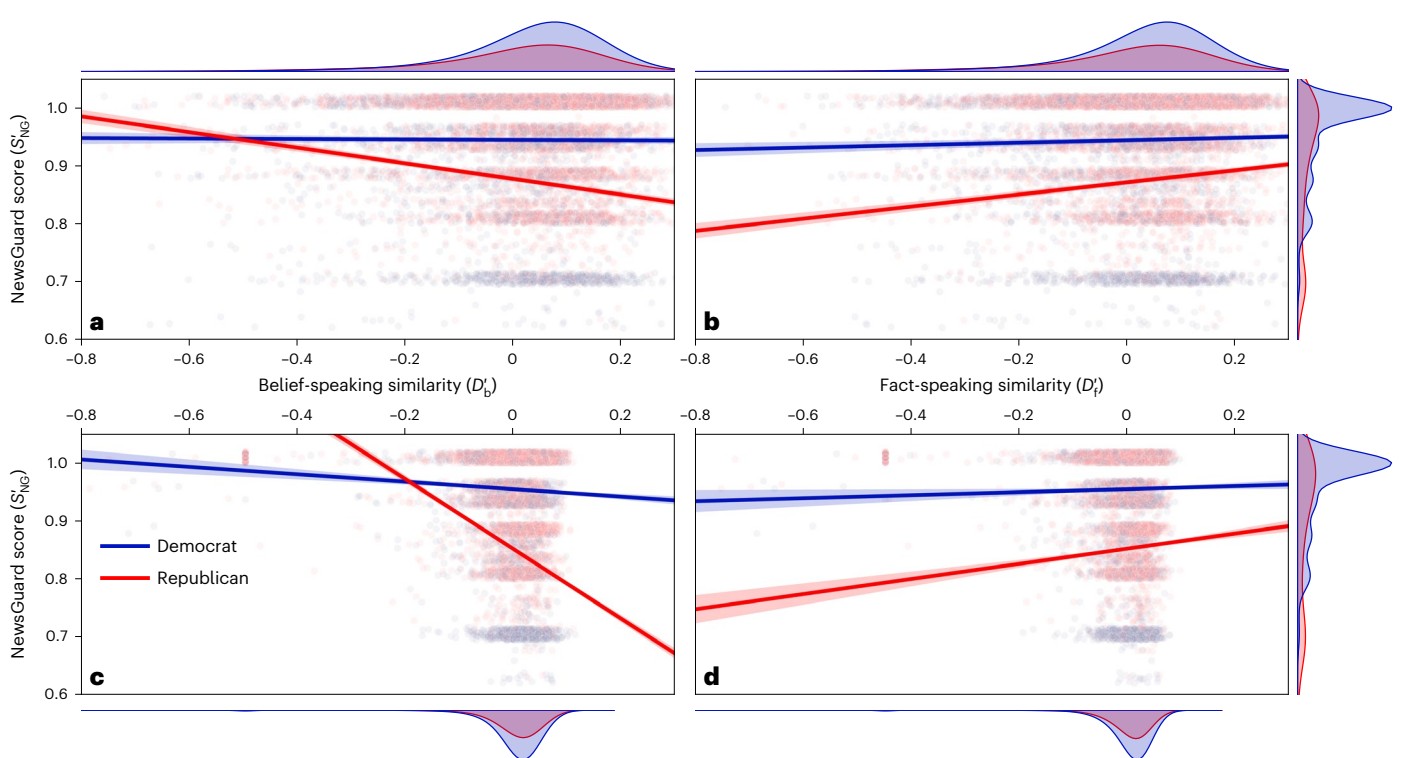

**Fig. 3 | Relation of information quality with belief speaking and fact speaking. a,b**, The rescaled NewsGuard score $S'_{NG}$ of links posted by individual US Congress members over belief-speaking ($D'_b$) (**a**) and fact-speaking ($D'_f$) (**b**) similarity measured in tweet texts, respectively. The lines and shaded areas indicate NewsGuard score predictions and 95% CIs from a linear mixed-effects model (equation (1)). **c,d**, The rescaled NewsGuard score $S'_{NG}$ over belief-speaking (**c**) and fact-speaking (**d**) similarity measured in article texts scraped from the tweeted links. The lines and shaded areas indicate NewsGuard score predictions and 95% CIs from a linear-regression model (equation (2)). The scatterplots show only $10^5$ data points per panel and vertical jitter was applied to visually separate data points. Note that we truncated the *y* axis at 0.6. The full data are shown in Extended Data Fig. 2. Marginal distributions on the sides show the kernel density estimation over the full data on the respective axes, separated by party.

state and party. We find that the quality of information being shared by Republicans tends to be lower in southern states (for example, AL, TN, TX, OK and KY) than in the north (for example, NH, AK and ME), although there are also striking exceptions (for example, NY). For Democrats, no clearly discernible pattern across states emerges.

We also find that the voting patterns during the 2020 presidential election in their home state did not affect the quality of news being shared by members of Congress.

To exclude a dependence of these results on use of the NewsGuard database, we validated this analysis with an independently collected

list of news-outlet reliability from academic and fact-checking sources. The results are reported in Supplementary Note 7 and are consistent with the results reported here. In addition, using the different outlet-reliability database, we also find a significant effect of belief-speaking similarity on the quality of shared information for Democrats that goes in the same direction as the effect for Republicans.

We wanted to know whether the content of belief-speaking and fact-speaking words in the texts found in the websites linked to be the tweets was also indicative of low information quality. To this end, we attempted to scrape the text of all linked websites (Methods). We successfully collected text from about 65% of links. We excluded texts with less than 100 words and only retained one copy of the text in the case when multiple tweets contained links to the same website. In addition, we excluded all articles collected from links that were posted by members of both parties (2,462 texts; 0.91% of articles), such that each link had a unique party designation. This resulted in a total of 261,765 unique news texts.

We investigated the dependence of the NewsGuard score associated with the domain from which the text was scraped on the belief-speaking similarity and the fact-speaking similarity of the article text (rather than in the original tweet). We fitted a linear-regression model to predict the rescaled NewsGuard score $S'_{NG}$ depending on party, the belief-speaking and fact-speaking similarities $D'_b$ and $D'_f$ and the two-way interaction terms (see equation (2) and Methods for details).

We show both the data for individual links and the model predictions for $D'_b$ and $D'_f$ in Fig. 3c and Fig. 3d, respectively. Again, we found a significant inverse relationship between $P$ = Republican and $S'_{NG}$ ($t$(261, 765) = −184.8, $P < 0.001$, coefficient −0.099, 95% CI = [−0.100, −0.098]) and the interaction term between Republican and $D'_b$ ($t$(261, 765) = −33.8, $P < 0.001$, coefficient −0.540, 95% CI = [−0.571, −0.509]). We also confirmed the positive relationship between $S'_{NG}$ and $D'_f$ ($t$(261, 765) = 2.3, $P = 0.003$, coefficient 0.026, 95% CI = [0.004, 0.048]), and the interaction term between Republican and $D'_f$ ($t$(261, 765) = 6.0, $P < 0.001$, coefficient 0.110, 95% CI = [0.074, 0.147]), and the three-way interaction term party $\times D'_b \times D'_f$ ($t$(261, 765) = −14.8, $P < 0.001$, coefficient −0.594, 95% CI = [−0.673, −0.516]).

In contrast to the analysis using tweet texts, we also find a significant negative relationship for $D'_b$ for Democrats ($t$(261, 765) = −6.6, $P < 0.001$, coefficient −0.065, 95% CI = [−0.084, −0.046]) and a significant interaction term $D'_b \times D'_f$ ($t$(261, 765) = 2.8, $P = 0.006$, coefficient 0.067, 95% CI = [0.019, 0.113]). See Extended Data Table 3 for the full regression statistics. Our analysis of article texts therefore reproduces the main results from our analysis of tweet texts.

## Discussion

We curated two dictionaries that captured the distinction between an evidence-based conception of honesty (fact speaking) and a conception based on intuition, subjective impressions and feelings (belief speaking). We confirmed the validity and diagnosticity of the dictionaries by soliciting ratings from human participants both for individual keywords and for documents, and by showing that belief speaking prevailed in opinion pieces in the *NYT* but not in their science section, whereas the reverse occurred for fact speaking.

Applying those dictionaries to public political discourse by members of the US Congress, represented by their tweets, we find a bipartisan increase of the use of both fact-speaking and belief-speaking language over time, in particular from late 2016 onward. The use of fact-speaking and belief-speaking language is particularly intense for controversial topics, and this is also a bipartisan phenomenon.

The parties differ considerably, however, when the quality of information being shared is considered. Overall Republicans tend to share information of lower quality than Democrats (see also ref. 42), and this difference is associated to belief speaking: the more Republicans engage in belief speaking, the more likely they are to share low-quality

information. There is no evidence (or little evidence; Supplementary Note 7) for this relationship for Democrats.

Our results have several theoretical and practical implications that deserve to be explored. First, our data cast a new light on several recent analyses of the US public's information diet that have shown that conservatives are more likely to encounter and share untrustworthy information than their counterparts on the political left[39,42–44]. Several reasons have been put forward for this apparent asymmetry, for example, that partisans are motivated to share derogatory content towards the political outgroup[45]. As greater negativity towards Democrats is mostly found in lower-quality outlets, conservatives may disproportionately share untrustworthy information because it is satisfying a need for outgroup derogation[46].

Our analysis offers another potential explanation, namely that the public may be sensitive to cues provided by the political elites that, as we have shown here, also differ considerably in the accuracy of content that they share on social media. Specifically, Republican politicians frequently, although not always, share low-quality information and are thus providing a cue to their partisan followers of the legitimacy of those outlets. Evidence for the sensitivity of the public to leadership cues has been observed in the climate-change arena, where the growing polarization of the public along party lines mainly resulted from the Republican leadership gradually assuming a more hostile stance towards the science of climate change[47].

Furthermore, our analysis provides evidence that belief speaking could be a 'gateway' rhetorical technique for the sharing of low-quality information. The more Republican politicians appeal to beliefs and intuitions, rather than evidence, the more likely they are to share low-quality information. For Democrats, this association was absent in the main analysis using NewsGuard scores, and it was attenuated if an independent source of domain quality was used (Supplementary Note 7). This pattern gives rise to the question why, if belief speaking gives licence to the sharing of misinformation, is it only Republicans (or mainly Republicans) who avail themselves of that option?

A possible answer can be found in the finding that belief speaking is associated with greater negative emotion (Supplementary Note 4). Therefore, belief speaking may result from Republican politicians' desire to derogate Democrats, as suggested in ref. 46. With this view, negative emotional content should be a mediator of the association between belief speaking and low quality of shared content. In contrast, if belief speaking were instrumental in the sharing of low-quality content for other reasons, then it should mediate the association involving negative emotionality. We report two competing mediation models in Supplementary Note 10. Although the models cannot definitively adjudicate between the two possibilities, the analyses suggest the former hypothesis is in a better position to explain the mediating effect on the spread of low-quality news among Republicans. Within this framework, and concordant with ref. 46, negative emotion associated with derogation of the opponent is the driving force behind the association between belief speaking and the spread of low-quality content among Republicans. Further indirect support for this possibility is provided by the fact that Republican members of Congress do not exclusively share misinformation. When they engage in fact speaking, Republicans' accuracy of shared information rises to nearly the same level as that of Democrats.

Finally, we return to the argument advanced at the outset, namely that belief speaking can be a marker of authenticity, which allows partisan followers to consider a politician to be honest despite them promulgating low-quality or false information. We cannot directly test this argument based on the present data because we have no way of ascertaining the perceived honesty of the politicians in our sample. However, we do have state-level electoral data from the 2020 presidential election, which show that belief speaking and the associated sharing of low-quality information is not associated with an electoral penalty (Supplementary Note 9). There is no association between the accuracy

of Republicans' shared information and the vote share for Trump, suggesting that voters were not deterred by belief-speaking-based dissemination of misinformation.

A limitation of our study is that it does not provide causal evidence because the reported observations are purely correlational. In addition, our analysis was limited to communications by the 'political class' in the United States and, although the United States is one of the world's largest democracies, the trends shown here should not be considered in isolation but deserve to be contrasted with observations in other countries and cultures. A recent comparison of the overall accuracy of information shared by members of the US Congress found that their accuracy was lower—even among Democrats—than the information shared by parliamentarians from mainstream parties in the United Kingdom and Germany[42]. Although there were also differences between parties in those two countries, they were small in magnitude and European conservatives were more accurate than US Republicans, underscoring that conservatism is not, per se, necessarily associated with reliance on low-quality information. Another international comparison of populist leaders (Trump in the United States, Modi in India, Farage in the United Kingdom and Wilder in the Netherlands) found some commonalities among those politicians, such as the use of insults against political opponents, but also identified Trump as an outlier in the use of critical language[48]. Therefore, further examinations of belief speaking and fact speaking outside the US context are urgently needed to explore the generality of our findings and to redress the existing global imbalance in research activity[7].

Future research is also needed to examine the temporal stability of the patterns we observed here. Although our analysis extended to the end of 2022, thus covering 2 months of Twitter activity after it was taken over by Elon Musk, there is no guarantee that the platform will remain stable in the future. Likewise, in the same way that sharing of misinformation mushroomed after 2016 (ref. 42), the long-term trend towards populism may reverse and the sharing of misinformation may become less frequent in the future. Therefore, our analysis is best understood as a historical and contemporary picture of political discourse rather than as a pointer to the future.

Finally, future research should also address the particular role played by social media in our analysis. We de-emphasized this angle because when our analysis was extended to mainstream news articles shared by the members of Congress, we found very similar results compared with the tweets. However, there may be other situations in which social media play a uniquely different role from conventional mainstream media, and those situations remain to be identified and examined.

## Methods

### US Congress member tweet corpus

A corpus of contemporary political communication in English was created by scraping tweets by members of both houses of the US Congress on 10 February 2023. To build the corpus, lists of Twitter handles of members of Congress were collected for the 114th (from www.socialseer.com), 115th (from www.socialseer.com), 116th (from ref. 49), and 117th and 118th (from https://triagecancer.org/congressional-social-media) Congresses. For the 114th and 115th Congresses, only the Twitter handles of senators were available. For the 116th, 117th and 118th Congresses, Twitter handles were available for both houses of Congress. This resulted in a total of 1,278 unique Twitter handles, which included Congress member staff and Congress member campaign accounts. If a politician had multiple accounts, all were included in the dataset. No sampling was involved in collecting the data and the collected dataset is exhaustive.

For each of the Twitter handles, metadata were collected on 10 February 2023 via the Twitter application programming interface (API) v.2 using the Python package twarc (v.2.13.0) (ref. 50). Metadata included the account's handle, username, creation date, location, user

description, number of followers, number of accounts followed and tweet count. Of the 1,278 accounts, 220 were not accessible because they had been deleted, suspended or set to 'private'.

To build the text corpus, all tweets posted by the collected Twitter accounts starting from 6 November 2010 and up to 31 December 2022 were collected, using academic access to the Twitter API. Note that by following this approach, we include all tweets posted by a given account in the given time span, not only tweets that were posted while a politician was in office. Earlier tweets all the way back to 2006 could be retrieved, but we chose 2010 as the earliest date due to changes in the design of retweeting in the Twitter platform at that time. The retweet button was introduced in November 2009 (previously, retweeting was done manually), and it took approximately a year for users to start using it consistently. Furthermore, the prominence of Twitter in US politics emerged later, especially since 2012. The resulting corpus consisted of a total of 5,914,107 tweets, of which 3,463,409 were original tweets, 531,289 were quote tweets, 575,044 were replies and 1,351,346 were retweets. Note that quoting, replying and retweeting are not exclusive categories. We removed retweets from the corpus because they do not constitute original content. The number of tweets consistently increased from around 100,000 in 2011 to over 600,000 in 2020, and then declined to around 500,000 in 2022. We removed exact matches (that is, duplicates) and included only tweets with more than ten words. The final corpus contained 3,897,032 tweets. Next to the tweet text, the corpus contained the tweet creation date and a unique identifier of the account that posted the tweet. The identifier permitted linkage to the metadata collected about the user accounts, such as party affiliation.

We find a large variance in the number of tweets posted by individual accounts, ranging from only 1 tweet in the observed time period to 52,055 tweets, with a median number of 2,876 tweets per account. To exclude a dependence of our results on highly prolific accounts, we also conducted the main analysis reported in Fig. 3 and Extended Data Table 2 using only the latest 3,200 tweets per account. Results from this analysis are highly consistent with the analysis using all available tweets (see Supplementary Note 11 for details). In addition, we show which accounts contribute most to the overall increase of belief speaking and fact speaking (Supplementary Note 12).

In addition to the perspective from individual tweets taken in the analysis presented in 'Partisan and temporal dynamics of conceptions of honesty', we also considered the perspective of individual links taken in the analysis presented in 'Relating honesty components to information trustworthiness'. For this analysis, we only considered tweets that contained at least one link (2,700,539 tweets). As a single tweet can contain more than one link, we expanded the dataset such that each entry referred to a single link, transferring the tweet-level honesty-component labels to the individual links. This resulted in a total of 2,844,901 links. From each link, we extracted the domain to which the link pointed. If the link was shortened using a link-shortening service, such as bit.ly, we followed the link to retrieve the full domain name. The domains were then matched against the NewsGuard domain trustworthiness database and the independently compiled list of trustworthiness labels (described in 'NewsGuard nutrition labels' and Supplementary Note 7).

### Honesty-component keywords and validation

We relied on keywords to identify the relevant subsets of tweets that involved the presumed distinct conceptions of honesty. Initially, two lists of keywords, one for each honesty component, were generated by the researchers involved in this article. The aim was to capture linguistic cues whose presence might signal that one of the components has been enacted by the speaker. To illustrate, initial keywords for fact speaking included terms such as 'reality', 'assess' 'examine', 'evidence', 'fact', 'truth', 'proof' and so on. For belief speaking, initial keywords were terms such as 'believe', 'opinion', 'consider', 'feel', 'intuition' or 'common sense'.

The lists were expanded computationally using a combination of the fastText library[51] and colexification networks[52,53]. Using the fastText embeddings, we expanded the seed words to include words that have a cosine similarity score above 0.75. Colexification networks connect words in a language based on their common translations to other languages, thus signalling words that can be used to express multiple concepts. For example, the words 'air' and 'breath' are considered to be colexifications because they both translate into the same word in multiple languages ('sukdun' in Manchu, 'vu:jnas' in Kildin Sami and 'jind'in Nenets[54]). Colexification networks have been used recently to study emotion structures in language[55] and are predictors of word-meaning ratings[52]. Including colexification networks in lexicon expansion gives word lists with a better trade off between precision and recall[53] than previous approaches using WordNet or word embeddings, such as empath. We subsequently filtered the expanded lists to remove duplicates, overlapping terms appearing in more than one list and lemma inflections (that is, 'convey', 'conveys' and 'conveyed'). The keywords were then used to identify texts relevant to the presumed conceptions of honesty.

To validate the keyword lists, we asked participants in an online survey to score each term on two scales reflecting the honesty components. Data were acquired on 20 September 2022 from 50 individuals (15 men, 34 women and 1 unlisted; mean age 39.5 years, s.d. 15.8 years) using the Prolific survey platform[56]. Participants were asked to score each term on two distinct Likert scales ranging from one to five, which indicated low and high representativeness, respectively, of the word for that honesty component. The instructions provided to participants can be found in Supplementary Note 1. The distributions of ratings collected for each keyword are shown in Supplementary Figs. 1 and 2.

We next performed paired $t$-tests to see how participants sorted the terms into the two conceptions. The results of the $t$-tests are shown in Supplementary Table 1. Of 98 keywords, 61 were judged to belong to the category to which we previously assigned them, 24 did not reach the significance threshold ($P < 0.05$) and were therefore removed and 13 were classified by participants as belonging to the opposite category. We followed the raters' indications and moved the keywords that were classified as belonging to the opposite category from their original dictionary to the other dictionary. The final list of keywords for both dictionaries is given in Extended Data Table 1.

### Identification of honesty components in text

As a first preparatory step, we removed URLs and replaced user handles on Twitter with the word 'user'. We then split the tweet texts into individual tokens (words). We then created embeddings of each word contained in the honesty-component dictionaries (Extended Data Table 1) with GloVe[57] trained on 840B tokens from the Common Crawl corpus, following the distributed dictionary representation approach[31]. We note that the word 'seem' from the belief-speaking dictionary is included in the list of stopwords of GloVe. Therefore, we removed seem from the stopword list to include it into the dictionary embedding that was calculated using GloVe.

We then averaged the single-word embeddings within every honesty component to create an embedded representation of the entire dictionary. Similarly, we embedded every token contained in a given tweet and calculated an average of all token embeddings to create an embedded representation of the tweet. For every tweet and both components, we then calculated the cosine similarity between the embedded tweet representation and the embedded dictionary representations to arrive at a belief-speaking similarity score $D_b$ and a fact-speaking similarity score $D_f$ for the given tweet. Similarity scores range from −1 (not similar at all) to 1 (perfectly similar).

We find that similarity scores correlate with the length of tweets (number of characters), with Pearson's $r = 0.37$ ($P < 0.001$) for belief speaking and $r = 0.42$ ($P < 0.001$) for fact speaking. In addition, the length of tweets systematically increases over the years, particularly after the increase in the tweet character limit from 140 characters to 280 characters in 2017. To remove the trend in similarity scores due to increasing tweet length, we fit two linear models $D_b$ ~ tweet length and $D_f$ ~ tweet length. We then used these linear models to predict $D_b$ and $D_f$ for every tweet based on its length and subtracted this prediction from the measured belief-speaking and fact-speaking similarity, resulting in the centred and length-corrected similarity scores $D'_b$ and $D'_f$, which we report throughout this article.

To measure belief-speaking and fact-speaking similarity in the text of the articles collected from links posted by Congress members on Twitter (see 'News article collection'), we followed the same approach as described for the text of the tweets above but measure the length of an article as the number of words it contains instead of the number of characters.

To test the robustness of our results to perturbations of the dictionaries, we recalculated belief-speaking and fact-speaking similarities using versions of the dictionaries where 7 words (20%) were removed from the dictionary at random before embedding the words and calculating dictionary representations. We then re-ran the regression of $S'_{NG}$ on $D'_b$, $D'_f$, party and the interaction terms (equation (1)), where $D'_b$ and $D'_f$ are the belief-speaking and fact-speaking similarities calculated using the representations of the perturbed dictionaries. The distribution of estimates for the fixed effects of the two-way interaction between party and $D'_b$, and party and $D'_f$ over 100 perturbations are shown in Extended Data Fig. 4. Whereas the estimates for the effect of $D'_b$ and $D'_f$ on NewsGuard score vary by about 20% between different perturbed dictionary versions, the effects never change direction and always remain significant ($P < 0.001$) for Republicans, as reported in the main text.

In addition to GloVe[57] embeddings, we also calculated $D'_b$ and $D'_f$ using word2vec[58] and fastText[51] embeddings of both the dictionary keywords and the tweets to exclude a dependence of our results on the choice of embedding. We note that, similar to GloVe, the word seem is included in the stopword list of word2vec and was removed from the stopword list before computing the embeddings. The results of fitting the linear mixed-effects model following equation (1) using the alternative embeddings for the dictionaries and tweet texts are shown in Supplementary Note 13. The results are similar to the results obtained using GloVe embeddings (Extended Data Table 2). This shows that our results do not depend on the algorithm or the corpus (common crawl for GloVe and word2vec versus Google news for fastText) that was used to train the embedding.

Lastly, we also investigated which individual keyword probably contributed the most to the overall increases of belief speaking and fact speaking reported in Fig. 2. We report the results in Supplementary Note 14.

### Honesty-component document-level validation

To validate our measures of the belief-speaking and fact-speaking honesty components on the document level, we asked human raters to rate individual tweets with respect to their similarity to the two honesty components. To this end, we sampled 20 tweets from the top belief-speaking and bottom fact-speaking quartile, and 20 tweets from the top fact-speaking and bottom belief-speaking quartile. In addition, we sampled 20 tweets that simultaneously belonged to the bottom belief-speaking and fact-speaking quartiles. Each sample of 20 tweets included 10 tweets from Democrats and 10 from Republicans.

We then created a survey on Prolific[56] and asked participants ($N = 51$) to rate each tweet's representativeness of the two honesty components on two separate Likert scales. We followed exactly the same set-up as described in 'Honesty component keywords and validation' above, but presented full tweets instead of singular keywords. The instructions provided to participants can be found in Supplementary Note 1. In addition, we included an attention check in the survey, with the aim of excluding all participants that failed the check. To this

end, we asked all participants to select '5' for both categories halfway through the survey. Only one person failed the check. The responses of this person were excluded from the survey, resulting in $N = 50$ total responses (25 men, 24 women and 1 non-binary person; mean age 37.6 years, s.d. 12.88 years). Data were acquired on 10 February 2023. The distributions of ratings collected for each tweet are shown in Supplementary Note 2.

We then wanted to quantify the performance of our computed similarity scores when used as a classifier. To this end, for each honesty component we coded the 20 tweets that were selected from the top belief-speaking (fact speaking) similarity quartile as belief speaking (fact speaking) and the 40 tweets that were selected from the bottom similarity quartile of that component as 'not belief speaking' ('not fact speaking'). We then classified every tweet for which a majority of human raters selected either a four or a five for how characteristic a tweet was for belief speaking (fact speaking) as belief speaking (fact speaking) to create a ground-truth dataset to compare our classifier against. We obtained receiver operating characteristic curves for belief speaking and fact speaking by varying the threshold for belief-speaking (fact speaking) similarity to categorize a tweet as belief speaking (fact speaking) (akin to varying-response criteria in a behavioural study). The receiver operating characteristic curves are shown in Supplementary Note 3. The area under the curve is high in both cases, with AUC = 0.824 for belief speaking and AUC = 0.772 for fact speaking.

### NYT corpus

We retrieved data from the *NYT* through their archive API (https://developer.nytimes.com/docs/archive-product/1/overview). By iterating over the months since the founding of the newspaper in the 19th century, we retrieved information on every article in the archive. The information returned by the API included the article title, an abstract that summarizes the article content, and additional metadata such as publication date and section of the paper. This approach is different to earlier research that used the *NYT* API to obtain a number of articles over time that contain certain terms, which does not yield any further text or ways to filter the data[59]. As we needed text to identify honesty components in articles, the archive end point was more suitable than the term search function of the *NYT* API, despite not giving us the full text of all articles but only returning a summary.

We extracted three distinct categories of content from the *NYT* corpus based on the sections identified in the metadata: (1) an 'opinion' category that comprises opinion pieces such as 'op-eds'; (2) a 'politics' category consisting of articles in the sections United States, Washington and world; and (3) a 'science' category that includes health, science, education and climate articles. We chose these three clusters because we expected opinion articles to contain more belief speaking, whereas we expected science articles to contain more fact speaking. We expected articles in the politics cluster to fall in between. We retrieved a total of 809,271 articles consisting of 240,567 opinion articles, 518,123 politics articles and 50,581 science articles.

### Word and topic keyness analysis

The scatterplot in the top panel of Fig. 1 was produced using the approach described in Scattertext[37], a Python package designed to illustrate words and phrases that are more characteristic of a category, such as political party, than others. To derive how characteristic a word is of a category, we start from raw word frequencies: for each word $w_i \in W$ and category $c_j \in C$, we define the precision of the word $w_i$ with respect to the category as

$$\text{prec}(i,j) = \frac{\#(w_i, c_j)}{\sum_{c \in C} \#(w_i, c)}.$$

Here the function $\#(w_i, c_j)$ represents the number of times $w_i$ occurs in a document labelled with the category $c_j$. Therefore, prec$(i,j)$ represents

the discriminative power of a given word across categories regardless of its frequency in the given category.

Similarly, we define the frequency a word occurs in a category $c_j$ as

$$\text{freq}(i,j) = \frac{\#(w_i, c_j)}{\sum_{w \in W} \#(w, c_j)}.$$

To combine prec$(i,j)$ and freq$(i,j)$ into a single score, we scale and standardize both values using a normal cumulative density function $\Phi(z)$ and then calculate the harmonic mean (H) between the two contributions (see ref. 37 for details). This yields the scaled *F*-score (SFS) for every word $w_i$ and category $c_j$ that is defined as

$$\text{SFS}(i,j) = \mathcal{H}\left(\Phi(\text{prec}(i,j)), \Phi(\text{freq}(i,j))\right).$$

For our application case, we want to show how representative a word is not only for a single category (such as 'Republican') but rather on a spectrum of representativeness that ranges from 'more Democratic' to 'more Republican'. To this end, we need to map the two distinct scores $\text{SFS}^D$ for the category Democratic and $\text{SFS}^R$ for the category Republican to a single score that ranges from −1 to +1. For two arbitrary categories $x$ and $y$ we therefore define

$$\text{SFS} = 2 \times \left( -0.5 + \begin{cases} \text{SFS}^x & \text{if SFS}^x > \text{SFS}^y, \\ 1 - \text{SFS}^y & \text{if SFS}^x < \text{SFS}^y, \\ 0 & \text{otherwise} \end{cases} \right).$$

This maps two SFSs (one for category $x$ and one for category $y$) that are both defined in the range [0, 1] to a single score in the range [−1, 1]. To this end, $\text{SFS}^y$ is mapped to [−1, 0], the SFS with the larger magnitude is selected and is then rescaled to the new range. In our application case, this then yields a single $\text{SFS}^{\text{party}}$ that is −1 for more Republican tweets and +1 for more Democratic tweets.

To calculate representativeness along the belief-speaking–fact-speaking dimension, we follow a similar approach. Before we can calculate the SFS for belief speaking and fact speaking, we first need to transform the continuous honesty similarity scores $D'_b$ and $D'_f$ into a binary honesty-component label for each tweet. To this end, we divided the tweets into quantiles according to their belief-speaking (fact speaking) similarity. We then categorized the tweets with a belief-speaking (fact speaking) similarity in the top 20% as belief speaking (fact speaking). If a tweet was part of the upper quantile for both components, then the higher of the two similarity values was used to assign a category to the tweet. We then followed the approach described above to calculate a single $\text{SFS}^{\text{honesty}}$ from $\text{SFS}^b$ (for belief speaking) and $\text{SFS}^f$ (for fact speaking).

As a result, each word had two SFS scores: $\text{SFS}^{\text{party}}$ and $\text{SFS}^{\text{honesty}}$. These two scores were used as $x$ and $y$ coordinates for the scatterplot shown in the top panel of Fig. 1. The X-shaped structure of the words in the scatterplot indicates that words that are characteristic for one dimension (for example, party) are probably also characteristic for the other dimension (for example, honesty component). Words that are not characteristic of any category (such as stopwords) cluster in the middle.

### NewsGuard nutrition labels

Following precedent[39,40], we used source trustworthiness as an estimator for the trustworthiness of an individual piece of shared information. We used nutrition labels provided by NewsGuard, a company that offers professional fact checking as a service and curates a large database of domains. The trustworthiness of a domain is assessed in nine categories, each of which awards a number of points: does not repeatedly publish false content (up to 22 points); gathers and presents information responsibly (18); regularly corrects or clarifies errors (12.5); handles the difference between news and opinion responsibly

(12.5); avoids deceptive headlines (10); website discloses ownership and financing (7.5); clearly labels advertising (7.5); reveals who is in charge, including any possible conflicts of interest (5); and the site provides names of content creators, along with either contact or biographical information (5).

NewsGuard categorizes domains with a score of 60 or higher as 'generally adheres to basic standards of credibility and transparency'[41]. Similar to ref. 60, we used this value as a threshold below which we categorized a domain and the link pointing to it as not trustworthy.

After excluding links to other social media platforms (for example, Twitter, Facebook, YouTube and Instagram) and links to search services (Google and Yahoo), the NewsGuard database covers between 20% and 60% of the links posted by members of the US Congress, with a steadily increasing share of links covered over time (Extended Data Figure 1a).

## Regression

We performed a range of regression analyses to quantify the relationship between various manifestations of honesty components and information quality. For the predictions shown in Fig. 3a,b, we fitted the following linear mixed-effects model for tweets from members of the US Congress:

$$S'_{NG} \sim 1 + D'_b \times D'_f + D'_b \times D'_f \times P + \left(1 + D'_b \times D'_f | \text{userID}\right) \quad (1)$$

Here, $S'_{NG}$ is the NewsGuard nutrition score of a domain to which a Congress member linked in a post on Twitter, rescaled to [0; 1]. $D'_b$ and $D'_f$ are the centred and length-corrected belief-speaking and fact-speaking similarity of the text in the tweet with the link, respectively (see section 'Identification of honesty components in text' above). $P$ is the party designation of the account that posted the tweet, which can be Republican or Democrat. We include random slopes and intercepts for every account (userID). We fitted the model using the lmer function from the R library lme4 (ref. 61). Regression results are reported in Extended Data Table 2. Data distribution was assumed to be normal but this was not formally tested.

For the predictions shown in Fig. 3c,d, we fitted the following model for articles that were linked to by the US Congress members:

$$S'_{NG} \sim 1 + D'_b \times D'_f + D'_b \times D'_f \times P. \quad (2)$$

Here, $D'_b$ and $D'_f$ are the centred and length-corrected belief-speaking and fact-speaking similarity scores of the article text retrieved from the link. We fitted the model using an ordinary least-squares fitting approach from the Python package statsmodels[62]. Regression results are reported in Extended Data Table 3. Data distribution was assumed to be normal but this was not formally tested. Note that we did not fit a linear mixed-effects model for the statistical analysis of the articles because there is no clear nesting of articles within individual Twitter accounts as a single article can be linked to from multiple accounts.

## News article collection

Excluding links to other social media platforms (for example, Twitter, Facebook, YouTube and Instagram) and links to search services (Google and Yahoo), our corpus of tweets contained 1,027,050 unique links to news articles that were shared by members of Congress. Of these links, 462,853 pointed to sites that were indexed by the NewsGuard database (see 'NewsGuard nutrition labels'). We scraped the text of these sites using Newspaper3k (ref. 63), a Python package for scraping and curating news articles. Some links were broken, restricted or could not be scraped by the package. In addition, we removed all articles that contained less than 100 words or were shared only by independent politicians (that is, not Republican or Democrat). This resulted in 65% of total scraping coverage. When broken down by trustworthiness, the coverage for trustworthy links ($N = 291,143$) was 65%, and 82% for

untrustworthy links with a NewsGuard score < 60 ($N = 7,776$). We retained only one copy of each news article in case it was shared multiple times, and removed from the main analysis articles that were shared by members of more than one political party (that is, a link was shared either by Republicans or Democrats, but not both). This was done to ensure each article had only a single party designation such that our statistical analysis of articles was comparable to our statistical analysis of tweets. This resulted in the removal of 2,462 articles (0.91% of all remaining articles), which were analysed separately. To provide a marker for apparent bipartisan agreement, we plot the mean and s.d. of honesty-component similarity and $S'_{NG}$ for the articles shared by both parties (grey ellipses in Extended Data Fig. 2). Removing these articles left us with a corpus of 261,765 article texts.

The distribution of NewsGuard scores and the belief-speaking and fact-speaking similarity in each article is shown in Extended Data Fig. 2c,d.

## Inclusion and ethics

This study is based on publicly available archival Twitter data on US Members of Congress and their official staff and campaign accounts. Only public figures are analysed and only content that was not deleted by the time of data retrieval was considered. All US Members of Congress in curated Twitter account lists are included as long as their Twitter accounts were public by the retrieval data. We focused on the two major parties to have sufficient evidence for statistical analysis, and our results cannot be extended to independent members of congress or members of other parties besides the Democratic and the Republican parties.

## Reporting summary

Further information on research design is available in the Nature Portfolio Reporting Summary linked to this article.

## Data availability

The lists of Twitter handles of members of congress used to build the tweet corpus are available from www.socialseer.com (114th and 115th Congress), ref. 49 (116th Congress) and https://triagecancer.org/congressional-social-media (117th and 118th Congress). The tweet identifiers of the tweet texts and URLs of the articles analysed in this study are deposited in the Open Science Framework (OSF)[64]. Dictionaries of keywords associated with the different conceptions of honesty are deposited in the OSF[64]. The independently compiled list of domain accuracy and transparency scores is deposited on GitHub[42]. The NewsGuard database used to asses domain trustworthiness is commercially available from NewsGuard and cannot be shared publicly. Aggregated values for information trustworthiness and honesty components for tweets and articles used to produce all figures in this article are deposited in the OSF[64].

## Code availability

Python 3.9.1 and R 4.2 were used to collect the data and perform the data analysis presented in this study. Data collection and analysis code is available under MIT License in a GitHub repository[65].

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

## Acknowledgements

This report was partly funded by the Templeton Foundation through a grant awarded to Wake Forest University for the Honesty Project. S.L. was also supported by funding from the Humboldt Foundation in Germany, and S.L. and D.G. are beneficiaries of the ERC Advanced Grant PRODEMINFO (101020961). J.L. was supported by the Marie Skłodowska-Curie grant number 101026507. The funders had no role in study design, data collection and analysis, decision to publish or preparation of the article. We acknowledge Travis Coan for helpful feedback on the manuscript.

## Author contributions

S.L., D.G. and J.L. conceptualized the research. S.T.A., F.C., J.L. and A.S. developed the methodology and statistical models. F.C. performed the validation. J.L., S.T.A., A.S. and F.C. performed computational and statistical analyses. J.L. and S.T.A. collected and curated the data. J.L. prepared the visualizations. J.L. administrated the project. S.L. and D.G. acquired funding and supervised the project. S.L. and J.L. wrote the original draft of the article. All authors contributed to editing the original draft of the article.

## Competing interests

The authors declare no competing interests.

## Additional information

**Extended data** is available for this paper at https://doi.org/10.1038/s41562-023-01691-w.

**Correspondence and requests for materials** should be addressed to Stephan Lewandowsky.

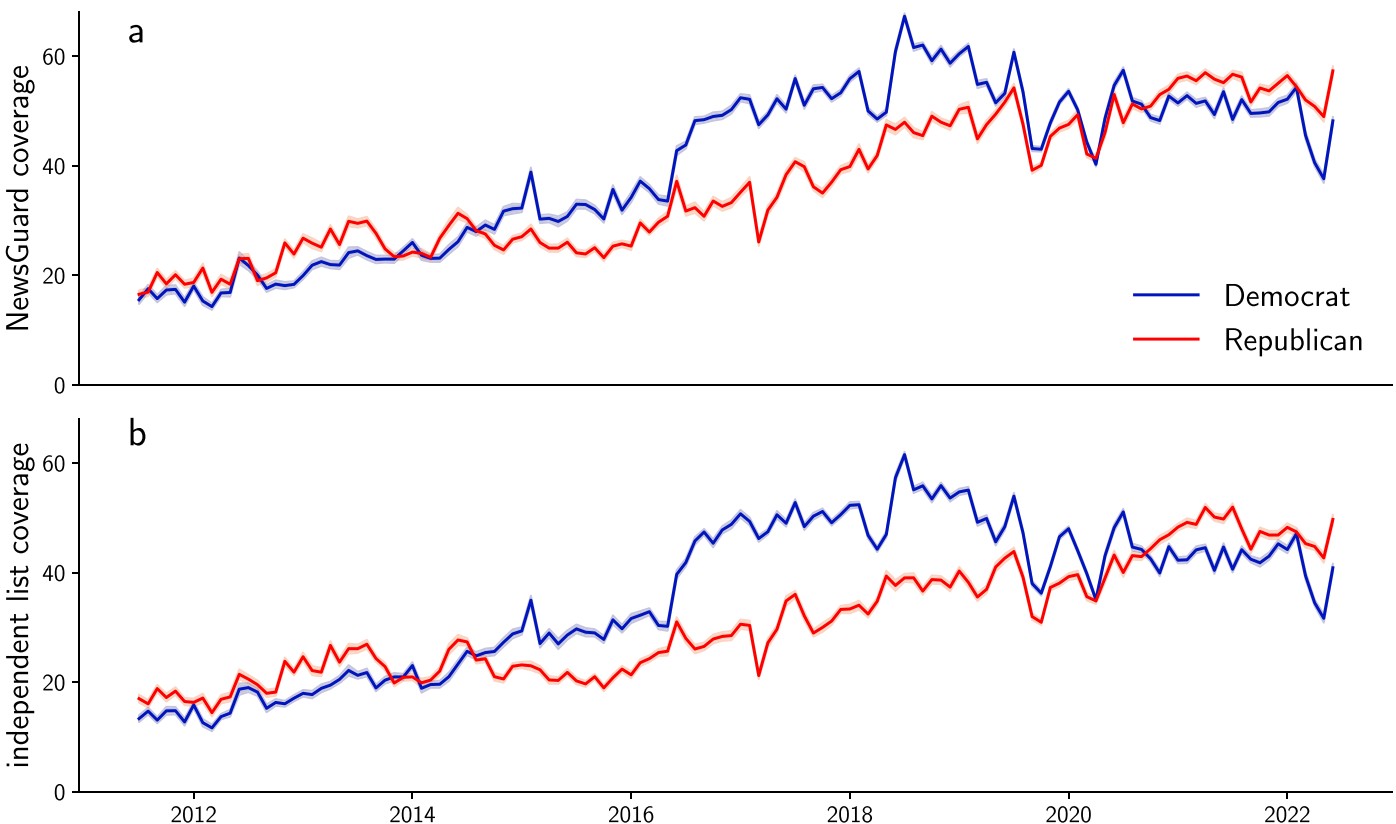

**Extended Data Fig. 1 | Share of links posted by accounts belonging to members of the U.S. Congress.** Share of links pointing to domains indexed in a the NewsGuard data base and b the independently compiled list.

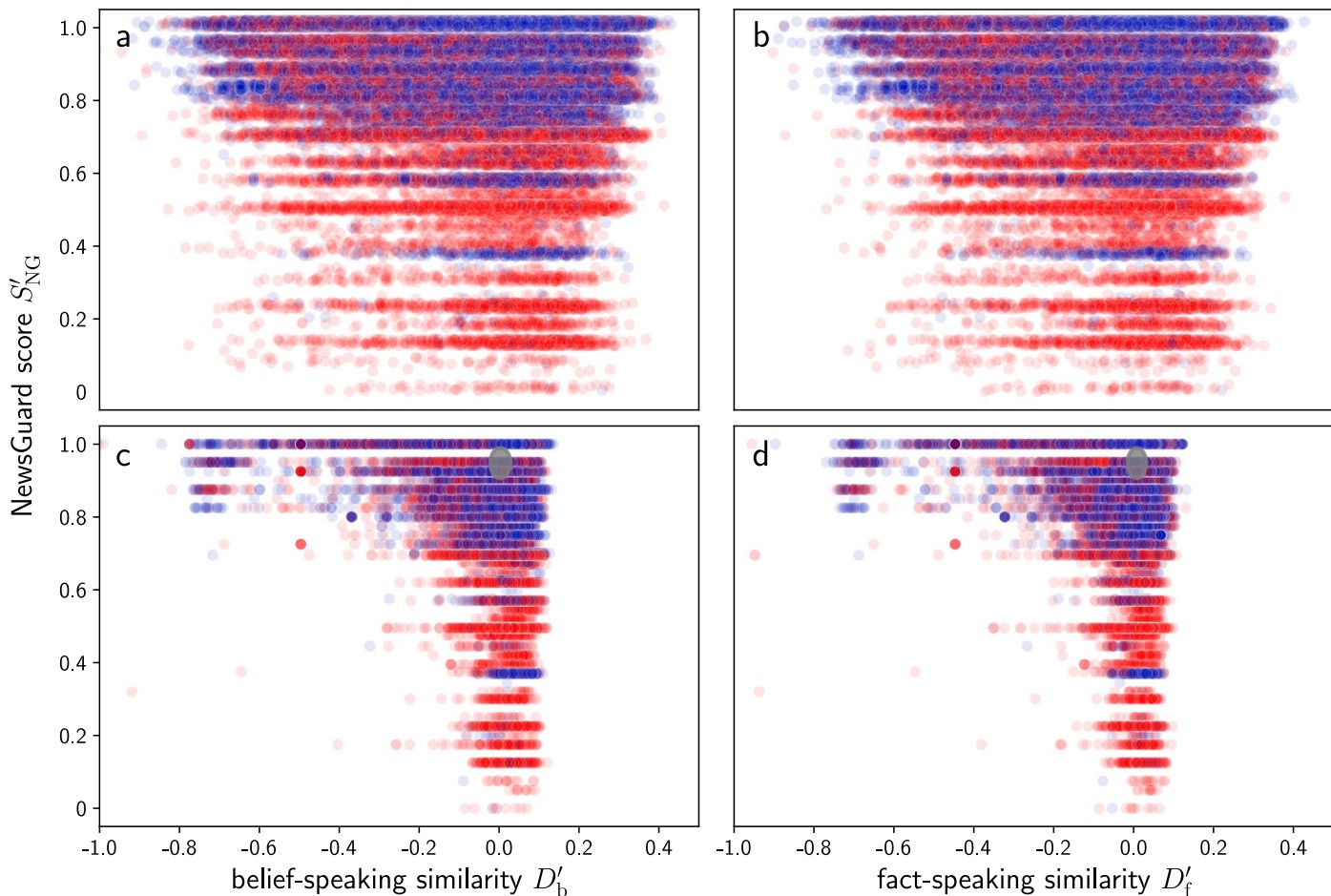

**Extended Data Fig. 2 | Relation of information quality with belief-speaking and fact-speaking (full data).** a and b rescaled NewsGuard score $S'_{NG}$ of links shared in tweets by members of the U.S. congress over belief-speaking similarity $D'_b$. Red and blue dots denote tweets by Democrats and Republicans, respectively. b shows $S'_{NG}$ over fact-speaking similarity $D'_f$ in tweets. c and d show the same information but with $D'_b$ and $D'_f$ calculated using the text of the articles that were linked instead of the tweet texts. The grey ellipses indicate the mean and standard deviation of the honesty component similarity and $S'_{NG}$ for articles shared by members of both parties. These articles were excluded in the regression analysis.

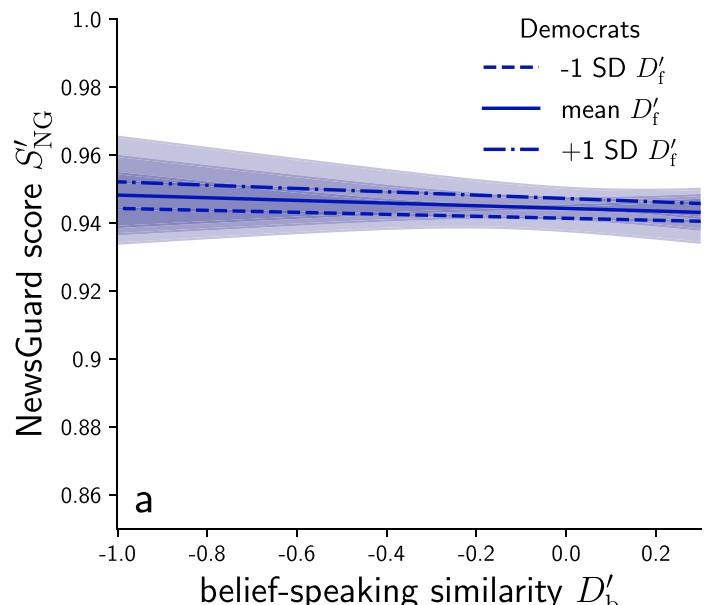

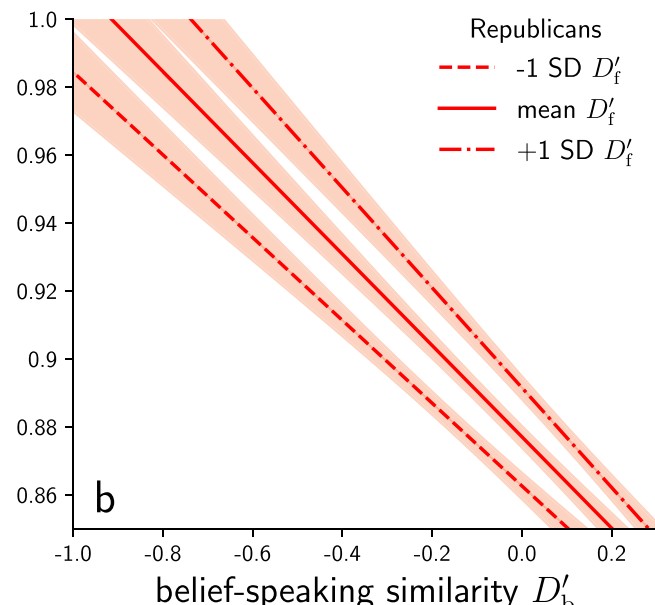

**Extended Data Fig. 3 | Three-way interaction between party, belief-speaking and fact-speaking.** Prediction of rescaled NewsGuard score $S'_{NG}$ for different values of belief-speaking similarity $D'_b$ and different levels (-1 SD, mean, +1 SD) of fact-speaking similarity $D'_f$ based on the fixed effect estimate of the three-way interaction $P \times D'_b \times D'_f$ (see linear mixed effects model in Eq. (1)) for tweets from **a** Democrat and **b** Republican members of the U.S. Congress.

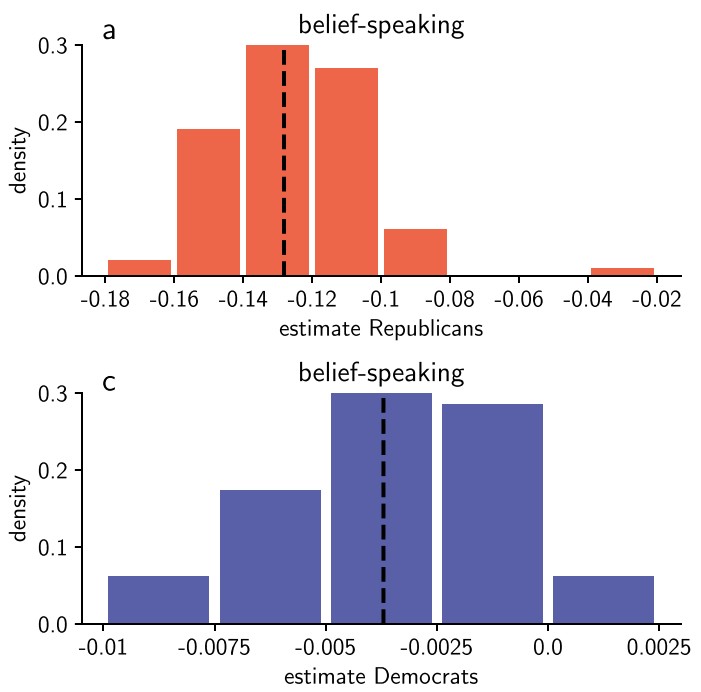

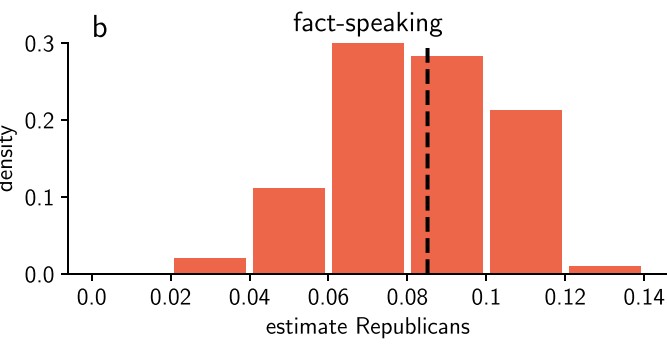

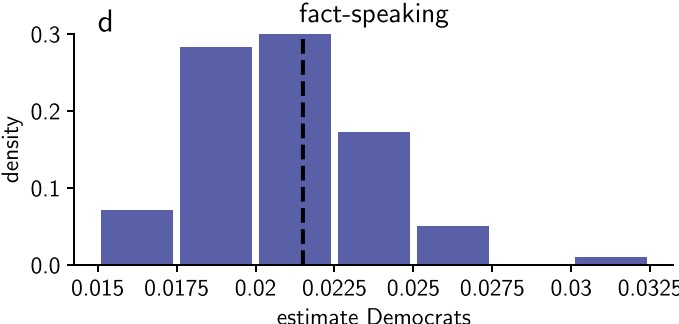

**Extended Data Fig. 4 | Dictionary robustness analysis.** a and b show the distribution of estimates of the effect of belief-speaking and fact-speaking similarity $D'_b$ and fact-speaking similarity $D'_f$ for Republicans from the linear mixed model (see Eq. (1)), where $D'_b$ and $D'_f$ were calculated with a perturbed dictionary for every tweet, respectively. c and d show the distribution of estimates of the effect of $D'_b$ and $D'_f$ for Democrats, respectively. Distributions were calculated from 100 dictionary perturbation iterations.

**Extended Data Table 1 | Keyword lists**

| Belief-speaking | fact-speaking |
|---|---|
| basically | actually |
| believe | analyze |
| claim | assess |
| confide | correct |
| consider | correction |
| contemplate | determine |
| contention | evaluate |
| envisage | evidence |
| feel | examine |
| frankly | exploration |
| genuinely | fact |
| guess | information |
| hint | inspect |
| judge | investigate |
| look | observe |
| obvious | proof |
| obviously | prove |
| of course | question |
| opinion | quiz |
| plainly | real |
| ponder | reality |
| position | rectify |
| presume | research |
| probably | revise |
| seem | sample |
| sensation | science |
| sentiment | scrutinize |
| signal | search |
| suggest | specify |
| suggestion | supervise |
| suppose | test |
| surely | trace |
| think | track |
| trust | trial |
| try | truth |
| view | validate |
| virtually | verify |

Lists of keywords for the two honesty components belief-speaking and fact-speaking.

**Extended Data Table 2 | Dependence of NewsGuard score on belief-speaking and fact-speaking measured in tweets**

|  | coef. | std. err. | $t$ | $P > |t|$ | [0.025 | 0.975] |
|---|---|---|---|---|---|---|
| Intercept | 0.9437 | 0.0016 | 582.104 | $< 10^{-16}$ | 0.9406 | 0.9469 |
| $D'_\mathrm{b}$ | -0.0037 | 0.0059 | -0.626 | 0.5317 | -0.0154 | 0.0079 |
| $D'_\mathrm{f}$ | 0.0215 | 0.0059 | 3.656 | 0.0003 | 0.0100 | 0.0330 |
| Republican | -0.0694 | 0.0023 | -29.894 | $< 10^{-16}$ | -0.0740 | -0.0649 |
| $D'_\mathrm{b} \times D'_\mathrm{f}$ | -0.0074 | 0.0099 | -0.741 | 0.4590 | -0.0268 | 0.0121 |
| $D'_\mathrm{b} \times$ Republican | -0.1282 | 0.0089 | -14.362 | $< 10^{-16}$ | -0.1457 | -0.1107 |
| $D'_\mathrm{f} \times$ Republican | 0.0851 | 0.0089 | 9.598 | $< 10^{-16}$ | 0.0677 | 0.1025 |
| $D'_\mathrm{b} \times D'_\mathrm{f} \times$ Republican | -0.0852 | 0.0151 | -5.645 | $2.6 \cdot 10^{-8}$ | -0.1148 | -0.0556 |
| Observations |  | 504809 | AIC |  |  | -800475 |
| Marginal $R^2$ |  | 0.086 | log-Likelihood |  |  | 400256 |
| Conditional $R^2$ |  | 0.182 | BIC |  |  | -800263 |

Results of a linear mixed effects model for the dependence of the rescaled NewsGuard score of each link $S'_\mathrm{NG}$ on belief-speaking similarity $D'_\mathrm{b}$ and fact-speaking similarity $D'_\mathrm{f}$ in tweets, with party $P$ as fixed variable following Eq. (1). The table reports results for the fixed effects. 504,809 observations were included. Regression was performed with the function lmer from the R library[51].

**Extended Data Table 3 | Dependence of NewsGuard score on belief-speaking and fact-speaking measured in articles**

| | coef. | std. err. | $t$ | $P > \lvert t \rvert$ | [0.025 | 0.975] |
|---|---|---|---|---|---|---|
| Intercept | 0.9556 | 0.0003 | 2904.316 | $< 10^{-16}$ | 0.9550 | 0.9563 |
| $D'_{\mathrm{b}}$ | -0.0651 | 0.0099 | -6.601 | $4.1 \cdot 10^{-11}$ | -0.0844 | -0.0457 |
| $D'_{\mathrm{f}}$ | 0.0257 | 0.0113 | 2.285 | 0.0229 | 0.0037 | 0.0477 |
| Republican | -0.0992 | 0.0006 | -184.756 | $< 10^{-16}$ | -0.1002 | -0.0981 |
| $D'_{\mathrm{b}} \times D'_{\mathrm{f}}$ | 0.0659 | 0.0239 | 2.757 | 0.0058 | 0.0190 | 0.1127 |
| $D'_{\mathrm{b}} \times$ Republican | -0.5400 | 0.0160 | -33.755 | $< 10^{-16}$ | -0.5713 | -0.5086 |
| $D'_{\mathrm{f}} \times$ Republican | 0.1104 | 0.0185 | 5.982 | $2.2 \cdot 10^{-9}$ | 0.0742 | 0.1465 |
| $D'_{\mathrm{b}} \times D'_{\mathrm{f}} \times$ Republican | -0.5943 | 0.0401 | -14.832 | $< 10^{-16}$ | -0.6728 | -0.5157 |
| R-squared | | 0.152 | Mean dependent var | | | 0.917 |
| Adjusted R-squared | | 0.152 | S.D. dependent var | | | 0.135 |
| Model MSE | | 103.5 | AIC | | | -348767 |
| Sum squared resid | | 4043 | BIC | | | -348683 |
| Log-likelihood | | 174391 | F-statistic | | | 6701 |
| Durbin-Watson stat | | 1.451 | Prob(F-statistic) | | | 0.000 |

Results of an ordinary least-squares regression for rescaled NewsGuard score of each link $S'_{\mathrm{NG}}$ on belief-speaking similarity $D'_{\mathrm{b}}$ and fact-speaking similarity $D'_{\mathrm{f}}$ in articles collected from links in tweets, following Eq. (2). 261,765 observations were included. Regression was performed with the function ols from the Python package[62], version 0.13.2.

# Reporting Summary

## Statistics

For all statistical analyses, confirm that the following items are present in the figure legend, table legend, main text, or Methods section.

| n/a | Confirmed | |
|---|---|---|
| ☐ | ☒ | The exact sample size (*n*) for each experimental group/condition, given as a discrete number and unit of measurement |
| ☐ | ☒ | A statement on whether measurements were taken from distinct samples or whether the same sample was measured repeatedly |
| ☐ | ☒ | The statistical test(s) used AND whether they are one- or two-sided *Only common tests should be described solely by name; describe more complex techniques in the Methods section.* |
| ☐ | ☒ | A description of all covariates tested |
| ☐ | ☒ | A description of any assumptions or corrections, such as tests of normality and adjustment for multiple comparisons |
| ☐ | ☒ | A full description of the statistical parameters including central tendency (e.g. means) or other basic estimates (e.g. regression coefficient) AND variation (e.g. standard deviation) or associated estimates of uncertainty (e.g. confidence intervals) |
| ☐ | ☒ | For null hypothesis testing, the test statistic (e.g. *F*, *t*, *r*) with confidence intervals, effect sizes, degrees of freedom and *P* value noted *Give P values as exact values whenever suitable.* |
| ☒ | ☐ | For Bayesian analysis, information on the choice of priors and Markov chain Monte Carlo settings |
| ☒ | ☐ | For hierarchical and complex designs, identification of the appropriate level for tests and full reporting of outcomes |
| ☐ | ☒ | Estimates of effect sizes (e.g. Cohen's *d*, Pearson's *r*), indicating how they were calculated |

*Our web collection on statistics for biologists contains articles on many of the points above.*

## Software and code

Policy information about availability of computer code

| Data collection | Data was collected with custom Python (v3.9.1) and R (v4.2) scripts. Specifically, the third-party Python packages twarc2 (v2.13.0) and newspaper3k (v0.2.8) were used. Code for data collection is available under accession code 10.5281/zenodo.7723109. |
|---|---|
| Data analysis | Data was analysed with custom Python (v3.9.1) and R (v4.2) scripts. Specifically, the third-party Python packages statsmodels (v0.13.2), scipy (v1.7.3), pengouin (v0.5.2), sentence transformers (v2.2.2) and torch (v1.8.1+cu102) and the third-party R package lme4 (v1.1-34) were used. Code for data analysis is available under accession code 10.5281/zenodo.7723109. |

For manuscripts utilizing custom algorithms or software that are central to the research but not yet described in published literature, software must be made available to editors and reviewers. We strongly encourage code deposition in a community repository (e.g. GitHub). See the Nature Portfolio guidelines for submitting code & software for further information.

## Data

Policy information about availability of data

All manuscripts must include a data availability statement. This statement should provide the following information, where applicable:
- Accession codes, unique identifiers, or web links for publicly available datasets
- A description of any restrictions on data availability
- For clinical datasets or third party data, please ensure that the statement adheres to our policy

The lists of Twitter handles of members of congress used to build the tweet corpus are available from www.socialseer.com (114th and 115th Congress), https://

doi.org/10.7910/DVN/MBOJNS (116th Congress), and https://triagecancer.org/congressional-social-media (117th and 118th Congress).The tweet IDs of the tweet texts and URLs of the articles analysed in this study are deposited in OSF under accession code https://doi.org/10.17605/OSF.IO/VNY8K.  Dictionaries of keywords associated with the different conceptions of honestyare deposited in OSF under accession code https://doi.org/10.17605/OSF.IO/VNY8K. The independently compiled list of domain accuracy and transparency scores is deposited on GitHub under accession code https://doi.org/10.5281/zenodo.6536692.The NewsGuard data base used to asses domain trustworthiness is commercially available from NewsGuard and cannot be shared publicly.Aggregated values for information trustworthiness and honesty components for tweets and articles used to produce all figures in this article are deposited in OSF under accession code https://doi.org/10.17605/OSF.IO/VNY8K.

# Research involving human participants, their data, or biological material

Policy information about studies with <u>human participants or human data</u>. See also policy information about <u>sex, gender (identity/presentation), and sexual orientation</u> and <u>race, ethnicity and racism</u>.

| | |
|---|---|
| Reporting on sex and gender | N/A |
| Reporting on race, ethnicity, or other socially relevant groupings | N/A |
| Population characteristics | N/A |
| Recruitment | N/A |
| Ethics oversight | N/A |

Note that full information on the approval of the study protocol must also be provided in the manuscript.

# Field-specific reporting

Please select the one below that is the best fit for your research. If you are not sure, read the appropriate sections before making your selection.

☐ Life sciences     ☒ Behavioural & social sciences     ☐ Ecological, evolutionary & environmental sciences

For a reference copy of the document with all sections, see <u>nature.com/documents/nr-reporting-summary-flat.pdf</u>

# Behavioural & social sciences study design

All studies must disclose on these points even when the disclosure is negative.

| | |
|---|---|
| Study description | Quantitative text analysis from social media and news. |
| Research sample | The Twitter accounts of U.S. Congress Members that were analysed in this study were compiled from public sources, e.g. www.socialseer.com (114th and 115th Congress), https://doi.org/10.7910/DVN/MBOJNS (116th Congress), and https://triagecancer.org/congressional-social-media (117th and 118th Congress). |
| Sampling strategy | The Twitter accounts included in this study are a comprehensive collection of Twitter accounts of U.S. Congress people that were active during the observation period 2011-2023. No sampling was involved in the compilation of Twitter accounts. |
| Data collection | Data retrieval through the Twitter API and The New York Times API. Analysis of the COHA corpus. |
| Timing | Twitter data was retrieved on February 12, 2023. The data spans a period between January 2011 and February 2023. |
| Data exclusions | Retweets, non-English tweets, duplicatet tweets were excluded. Only tweets from Democrat and Republican Congress Members were included. |
| Non-participation | US Congress Members who deleted their account or made it private prior to February 12, 2023 are self-excluded from the analysis. |
| Randomization | No randomization was performed, this is a purely observational study. |

# Reporting for specific materials, systems and methods

We require information from authors about some types of materials, experimental systems and methods used in many studies. Here, indicate whether each material, system or method listed is relevant to your study. If you are not sure if a list item applies to your research, read the appropriate section before selecting a response.

## Materials & experimental systems

| n/a | Involved in the study |
|---|---|
| ☒ | Antibodies |
| ☒ | Eukaryotic cell lines |
| ☒ | Palaeontology and archaeology |
| ☒ | Animals and other organisms |
| ☒ | Clinical data |
| ☒ | Dual use research of concern |
| ☒ | Plants |

## Methods

| n/a | Involved in the study |
|---|---|
| ☒ | ChIP-seq |
| ☒ | Flow cytometry |
| ☒ | MRI-based neuroimaging |

