## [Peer Review File · Nature Human Behaviour]

Peer Review Information

Journal: Nature Human Behaviour

Manuscript Title: From Alternative conceptions of honesty to alternative facts in communications by U.S. politicians

Corresponding author name(s): Stephan Lewandowsky

Reviewer Comments & Decisions:

Decision Letter, initial version:

18th August 2022

Dear Professor Lewandowsky,

Thank you once again for your manuscript, entitled "New conceptions of truth foster misinformation in online public political discourse," and for your patience during the peer review process.

Your manuscript has now been evaluated by 3 reviewers, whose comments are included at the end of this letter. In the light of their advice, I regret that we cannot offer to publish your manuscript in Nature Human Behaviour.

While the reviewers find your work of some interest, they raise concerns about the robustness of the data. We feel that these reservations are sufficiently important as to preclude publication of this work in Nature Human Behaviour.

I am sorry that we cannot be more positive on this occasion but hope that you will find our reviewers' comments helpful when preparing your paper for submission elsewhere.

Sincerely,

Samantha Antusch

Samantha Antusch, PhD
Editor
Nature Human Behaviour

Reviewer expertise:

Reviewer #1: truth-speaking ; misinformation ; Twitter data

Reviewer #2: polarization ; communication studies

Reviewer #3: SBERT ; NLP

Reviewers' Comments:

Reviewer #1:

Remarks to the Author:

Conceptual Novelty:

The work is novel, it uses a theoretical/classificational model from management studies and applies it to political speech. The model translates well to mainstream political discourse, and the computational methods (while still presenting some challenges due to the complexity and nuance of textual meaning) offer valuable potential for systemic analysis.

Advancement to Policy:

While authors see this work as being in the 'misinformation' space, I see this study as closer to political propaganda and polarization, and can be used to make sense of policy in that.

Methodological Advancement:

The main proposition of this work is to advance the methodology of capturing and analyzing political data, more on this in my detailed comments.

Key Contributions:

1. The mean share of belief-speaking and truth-speaking for both democrats and republicans has increased considerably, almost doubled in the period of the study, the authors argue this is related to the corresponding increase in the "fake news" narrative
2. Republicans are more likely to share belief-speaking tweets which reference sources that have low ratings for trustworthiness
3. An increase in truth-speaking corresponds to an increase in the average trustworthiness score of the sources tweeted by politicians, whereas an increase in belief-speaking corresponds to a (more significant) decrease in trustworthiness of sources quoted by politicians.
4. The paper brings a useful move away from misinformation broadly in binaries of truth and lie to the nuance of innuendo and opinion.
5. While variants of these have been reported in the mainstream news and in some literature, the results here are from a systematic study with a helpful conceptual model that has scope for replication

Some Suggestions for improvement:

1. It could help to have textbook examples of truth-seeking and belief-seeking tweets early in the text (instead of in the appendices for annotators)
2. Since these are public figures, authors could also consider a rank-ordered visualization of individual politicians or states (of the politicians) where either a truth- or belief-seeking approach is higher. While such analysis may help get at the contours of performative politics across states or issues. This ties in with a broader concern I have, in that the paper is largely written for computational data scientists, rather than for political scientists or scholars of discursive communication, who would benefit deeply from this work (and arguably would also be placed well to offer helpful critiques for improvement)
3. I was hoping some more analysis could be offered on Figure 1. While some results are intuitive (climate, discrimination), some are harder to make sense of (impeachment). Also is there any further analysis that can be offered on the distribution of frequencies? I found the death penalty pattern interesting, while authors point it out (republicans twice of democrats on truth and belief speaking) is there any further comment on that? Did authors look at the individual tweets on both sides to see what explains this?

4. While I can see why authors removed analysis articles that were shared by members of both parties, these could have benefitted from an eyeballing, both to check if there were partisan engagement here that crosses party lines.

Clarifications

1. Representatives and Senators often have multiple accounts (direct personal account, campaign account, senatorial account etc) -- how were these labeled. In the case of some representatives, they use their personal as their primary, whereas others use their senatorial/congressional alongside their personal accounts for different purposes. How were these reconciled and if only one account was used per politician, which was chosen and why?

2. I had some trouble parsing Extended Data Figure 2. First, it appears that the patterns for negative emotion, in particular, are very bursty, with sudden extreme dips followed by highs, including a period in 2014 where negative belief-speaking is extremely high among Democrats. I am not sure I read this right, but if so, what explains these?

3. On the inter-rater reliability test, the examples from the annotation guide felt like they were a bit difficult to parse, and it would be helpful to know if there was a debriefing with the raters on what they felt they were getting right or wrong once they reconciled their results with each other. Also, I am assuming that besides the basic classification into the three categories, the raters were judging if the tweet was in fact a negative-emotion belief-speaking.. and so on -- is this correct? Was the Krippendorff's α was computed on a binary agree/disagree (on say belief speaking... etc)? While I find the authors' argument for removing seeking-understanding acceptable, I'm still curious about what the authors feel about the lower values of the inter-rater tests, especially given the nature of sarcasm, innuendo, and indirect speech in political discourse. Also was inter-rater ground truth done following the LIWC positive/negative emotion scoring to confirm automated results?

4. Related to the above, do authors have any commentary on the incidence and quality of tweets that did the opposite of truth seeking while using the words (say for instance "proof", or "evidence" used as a discursive means to make belief-speaking seem more evidentiary)

References:

I would encourage authors to look beyond Euro/American research.

Two things come to mind. First, on the 'speaking their mind' track, one smaller data set approach the authors may find interesting is Gonawela et al (2018) "Speaking their Mind: Populist Style and Antagonistic Messaging in the Tweets of Donald Trump, Narendra Modi, Nigel Farage, and Geert Wilders"

Second, authors can also look at work that has done a two-party political discourse analysis using large scale electoral data, using hashtags in Panda et al Topical Focus of Political Campaigns and its Impact: Findings from Politicians' Hashtag Use during the 2019 Indian Elections

Reviewer #2:

Remarks to the Author:

Below I share my assessment of the study entitled "New conceptions of truth foster misinformation in online public political discourse", which was submitted to Nature Human Behaviour.

The study presents a thorough and insightful analysis, applies state-of-the-art methods, examines highly relevant issues, aims to contribute empirical findings to conceptual debates about conceptions of truth, evidence, and knowledge, integrates different disciplinary approaches from the social sciences (psychology, political communication, STS), and seeks to advance ongoing debates within and beyond the scholarly community about how misinformation, political communication, and partisan ideology relate to each other.

However, I have a number of criticisms and concerns. Some of these could be addressed in a substantial revision of the paper, but some weaknesses are, in my view, too severe to be fixed easily. These weaknesses decrease the overall quality of the paper substantially and outweigh the above-mentioned merits, which is why I cannot recommend publication in NHB.

In the following I provide detailed comments. Most of these revolve around the conceptual framing of the study, the interpretation of results, and the conclusion. I am not an expert in (automated) social media analyses, so I'll refrain a bit from commenting on the methodological and analytical procedures.

The conceptualisation of honesty vis-à-vis authenticity, veracity, accuracy etc.

There are a lot of terms and concepts in the paper's frontend which, in my view, are used somewhat interchangeably, such as "truth", "evidence", "accuracy", "honesty", "authenticity", or "veracity". Some of these terms are also (implicitly) opposed to "misinformation", "lying", and "evidence-free". I'd encourage the authors to disentangle these terms, which may mean explaining some of them briefly (particularly "misinformation", which was not defined although it is key to the present study), using them more consistently, or just using fewer different terms.

Relatedly, I am sorry to say that I was a bit confused by the authors' understanding of (dis)honesty: I can't see how honesty can be valued as "a signal of authenticity" (p. 2), also because the reference [8] does not say anything about (dis)honesty, rather about lying. Isn't it then rather that in some circumstances, people may even come to value *rejection of allegedly elitist evidence/truths* as a signal of authenticity? After all, Cooper et al.'s honesty model – even if useful for the present study – did not really help me much in understanding the concept of honesty nor in disentangling honesty from authenticity or accuracy, for example. Moreover, I am unsure if being honest can be considered an "establishment' norm" (p. 2). Some would not agree with this, because honesty can be conceived as a general virtue and is not precluded to a societal or political establishment.

The theoretical argument on (new) conceptions of truth

The authors argue that a "new ontology of truth" has emerged (p. 2). However, I am not sure if this ontology is actually new: "Alternative" or anti-hegemonic conceptions of truth and evidence – such as those analysed in the present study – have been present ever since (see van Zoonen, 2012, for example). What might be indeed new (at least for some contexts like the US) is an increasing salience of alternative truth conceptions in political discourse; not least due to social media, which help spreading and conveying salience to these conceptions, as they allow for circumvention of established gatekeepers, for example (Gerbaudo, 2018; Hopster, 2021). This assumption, however, has not been considered or discussed sufficiently, however. After all, the particularities of social media (and Twitter specifically) are barely discussed throughout the study, albeit the specificities of this communication environment are key to present study.

Relatedly, I think the authors might find a better title for the study, as I find the claim "conceptions [...] foster misinformation" somewhat unintelligible, also because it insinuates a causality that I don't think is supported by the results. Maybe it might be more accurate to use a title like "Misinformation links to alternative truth conceptions in online political discourse" or "Political misinformation and belief-speaking fuel each other" or something like this?

Relating truth-seeking to democracy and belief-speaking to fascism

This might be a minor point, but I'd be much more careful with implicitly equating the opposition of belief-speaking and truth-seeking with the opposition of fascism and democracy (see p. 3). Have belief-speaking ontologies really evolved out of fascist ideologies (as it is insinuated in "has roots in..")? Perhaps belief-speaking has been a feature or concomitant of fascist political thought, but I assume the same is true for some manifestations of nationalism, authoritarianism, or populism.

Introduction of research questions

While some aspects of the research questions (p. 4) have been well introduced and explained, other aspects lack a proper introduction, let alone a rationale for why the authors focused on them. For example, why did they choose to investigate US congress members? Do the US represent a special/characteristic or otherwise comparable/representative case? Why did the authors study partisan differences? Is there any reason to assume that they matter? (Sure there are, but the authors should at least briefly explain them)

Embedding in extant research

So far, only very few *empirical* studies on misinformation, political communication, partisanship, and social media have been cited. I encourage the authors to refer to some of these studies at least briefly in order to embed their findings and interpretations in the current literature.

Very low intercoder reliability

The authors find Krippendorff's Alpha values of .42 and .53 (with confidence intervals going as low as .30 and .31, respectively). This is, in my view, a substantial problem. The authors do suggest that Alpha values lower than .667 may be acceptable for ground-breaking research, citing Geiß (2021) and Riffe et al. (1998). But after having a closer look at these texts, I came to assume that $\text{Alpha} < .667$ is not acceptable even for novel and ground-breaking studies:

Geiß (2021) states:

"Krippendorff (2004a) suggests that Krippendorff's α 's (α 's) as high as .800 are necessary for trusting the coding, while α 's between .800 and .667 may suffice for drawing preliminary conclusions. Values lower than .667 are characterized as generally unacceptable. Riffe, Lacy, and Fico (1998) agree to the limits proposed by Krippendorff (2004a) but state that in exploratory and ground-breaking research lower values may suffice."

However, Riffe et al. (1998, p. 154) point out:

"However, Krippendorff (2004a) added that variables with Alphas as low as .667 could be acceptable for drawing tentative conclusions. The lower coefficient would be appropriate for research that is breaking new ground with concepts that are rich in analytical value."

These paragraphs might leave some space for speculation, but in my reading, this means that values lower than .667 are barely acceptable, even for novel research. Therefore I have strong concerns about the reliability of the analyses of the present study. When looking at the keyword lists for the three honesty components, my concerns intensified even more: I don't mean to nit-pick on single keywords – but I think actually a lot of them barely tap the honesty component that they mean to capture. For

example, “know”, “observe”, and “think” could well fit truth-seeking component, whereas “claim”, “tentative”, or “virtually” could also fit the belief-speaking component. Perhaps this is one of the reasons for the very low Krippendorff’s Alpha values.

Unequal lengths of keyword lists

I think it is indeed a notable limitation that the keyword lists for belief-speaking and truth-seeking differ. The authors do point out that this means that the frequency figures of each category may not be comparable. Moreover, however, I think this imbalance introduces further bias into the analyses: For example, all analyses on belief-speaking tweets may be less reliable/robust than the truth-speaking analyses – perhaps they may be considered underpowered in some sense. Also, the frequencies in Figures 1B and 1C are barely comparable because of this imbalance.

Interpretation and discussion of findings

The authors make several interesting and exciting findings – but barely discuss many of them. For example, they find that Republican truth-seeking keywords are linked to controversial matters or economic issues, whereas Democrat truth-seeking texts almost exclusively concern scientific topics (p. 6). The authors do not elaborate on this finding, although I think it is very relevant for the present study: To me it appears as if Republican truth-seeking is related to issues where there are no objective benchmark truths (or no endeavours by scientists/experts/etc to find such truths; e.g., tax, impeachment), whereas Democrat truth-seeking is indeed related to issues where there are consensual truths (e.g., climate). This suggests to me that Republican truth conceptions tend to rely on an understanding of truth that is rather related to practical questions of daily or societal life, whereas Democrat truth conceptions tend to rely on an understanding of truth that is related to epistemological questions.

Also, I’d encourage the authors to discuss the very interesting finding that approx. the same increase in belief-speaking tweets occurred for both parties, which is somewhat at odds with the popular narrative that only Republicans have come to promote their beliefs regardless of evidence-based truths in political discourse.

Coming to the conclusion section, I think it has substantial need for improvement and additions, for a number of reasons:

- Much of it is a summary of the findings
- It barely connects to the theoretical assumptions outlined in the introduction (e.g., the honesty model)
- It does not pick up the introductory argument on dishonesty (or criticisms of allegedly elitist knowledge claims) being an authenticity marker
- It does not reflect on the role of social media

- It does not discuss the specificity of the US context

All these additions would require more space, of course – which would be available if the authors, for example, report more briefly on the sensitivity analyses (NYT corpus and LIWC text analysis, p. 5-6). These analyses are very worthwhile, but I think they could be moved to the appendix or methods section, and only briefly be summarized in section 2.

References

- Geiß, S. (2021). Statistical Power in Content Analysis Designs: How Effect Size, Sample Size and Coding Accuracy Jointly Affect Hypothesis Testing – A Monte Carlo Simulation Approach. *Computational Communication Research*, 1(3). <https://doi.org/10.5117/CCR2021.1.003.GEISS>
- Gerbaudo, P. (2018). Social media and populism: An elective affinity? *Media, Culture & Society*, 40(5), 745–753. <https://doi.org/10.1177/0163443718772192>
- Hopster, J. (2021). Mutual affordances: The dynamics between social media and populism. *Media, Culture & Society*, 43(3), 551–560. <https://doi.org/10.1177/0163443720957889>
- Riffe, D., Fico, F., & Lacy, S. (1998). *Analyzing Media Messages: Using Quantitative Content Analysis in Research* (2nd ed.). Taylor and Francis.
- van Zoonen, L. (2012). I-Pistemology: Changing truth claims in popular and political culture. *European Journal of Communication*, 27(1), 56–67. <https://doi.org/10.1177/0267323112438808>

Reviewer #3:

Remarks to the Author:

This paper studies an important problem in the quality of political statements using Twitter data.

I have two major concerns:

- 1) The paper categorizes Twitter into belief speaking, truth-seeking, and others. How the categorization relates to misinformation is not clear. The paper uses NewsGuard to measure the correctness. However, the conclusion seems not conclusive based on Figure 3 with quite some differences between the two parties. Also, the usage of NewsGuard is not perfect as it is also an estimator.
- 2) The way to categorize the belief-speaking twitters and truth-seeking twitters is not scientific. The categorization is simply counting the number of word appearances. The results can be largely influenced by the selected keywords. The robustness is not clear which leads to untrustworthy results.

**Following suitable revisions, you may want to consider transferring your manuscript. I suggest that you consider Humanities and Social Sciences Communications as a suitable venue for your work. To transfer your manuscript there, please use our manuscript transfer portal [LINK REDACTED]. You will not have to re-supply manuscript metadata and files, unless you wish to make modifications, but please note that this link can only be used once and remains active until used. For more information, please see our manuscript transfer FAQ page.

Note that any decision to opt in to In Review at the original journal is not sent to the receiving journal on transfer. You can opt in to In Review at receiving journals that support this service by choosing to modify your manuscript on transfer. In Review is available for primary research manuscript types only.

Author Rebuttal to Initial comments

Response to reviewers

(Original text in `courier` font, our replies in *black italics*.)

Reviewer 1

Remarks to the Author:

Conceptual Novelty:

The work is novel, it uses a theoretical/classificational model from management studies and applies it to political speech. The model translates well to mainstream political discourse, and the computational methods (while still presenting some challenges due to the complexity and nuance of textual meaning) offer valuable potential for systemic analysis.

Advancement to Policy:

While authors see this work as being in the 'misinformation' space, I see this study as closer to political propaganda and polarization, and can be used to make sense of policy in that.

Methodological Advancement:

The main proposition of this work is to advance the methodology of capturing and analyzing political data, more on this in my detailed comments.

Key Contributions:

1. The mean share of belief-speaking and truth-speaking for both democrats and republicans has increased considerably, almost doubled in the period of the study, the authors argue this is related to the corresponding increase in the "fake news" narrative
2. Republicans are more likely to share belief-speaking tweets which reference sources that have low ratings for trustworthiness
3. An increase in truth-speaking corresponds to an increase in the average trustworthiness score of the sources tweeted by politicians, whereas an increase in belief-speaking corresponds to a (more significant) decrease in trustworthiness of sources quoted by politicians.
4. The paper brings a useful move away from misinformation broadly in binaries of truth and lie to the nuance of innuendo and opinion.
5. While variants of these have been reported in the mainstream news and in some literature, the results here are from a systematic study with a helpful conceptual model that has scope for replication

We appreciate the reviewer's thorough analysis and overall positive stance towards our work.

Some Suggestions for improvement:

1. It could help to have textbook examples of truth-seeking and belief-seeking tweets early in the text (instead of in the

appendices for annotators)

This is a great suggestion, and we now report examples of truth-seeking and belief-seeking tweets for both parties in Figure 1.

2. Since these are public figures, authors could also consider a rank-ordered visualization of individual politicians or states (of the politicians) where either a truth- or belief-seeking approach is higher. While such analysis may help get at the contours of performative politics across states or issues.

This is a very good point. We have conducted this analysis and report it in the online supplement (Section S8). We show that, for Republicans, it is predominantly southern states that attract lower NewsGuard ratings than states in the north, whereas for Democrats there is no readily discernible pattern.

This ties in with a broader concern I have, in that the paper is largely written for computational data scientists, rather than for political scientists or scholars of discursive communication, who would benefit deeply from this work (and arguably would also be placed well to offer helpful critiques for improvement)

We took note of the reviewer's concern and tried to broaden the appeal of the revision by emphasizing the implications for a broader community of scholars. It is difficult to measure this new slant, but we hope that the appeal of the paper is now broader.

3. I was hoping some more analysis could be offered on Figure 1.

We have redesigned Figure 1 so it focuses exclusively on the illustration of words, and we now also provide representative tweets to illustrate the four quadrants. We believe that this reduces complexity and illustrates the tenor of the messages better than the analyses of topics offered originally.

While some results are intuitive (climate, discrimination), some are harder to make sense of (impeachment). Also is there any further analysis that can be offered on the distribution of frequencies? I found the death penalty pattern interesting, while authors point it out (republicans twice of democrats on truth and belief speaking) is there any further comment on that? Did authors look at the individual tweets on both sides to see what explains this?

This part of the former Figure 1 is now in the online supplement (Section S4, Figure S5).

Owing to the conversion of the analysis from a simple keyword search to an embedding approach (see below), the figure now looks different and, hopefully, affords greater ease of interpretation.

4. While I can see why authors removed analysis articles that were shared by members of both parties, these could have benefitted from an eyeballing, both to check if there were partisan engagement here that crosses party lines.

In response, we have now also included an analysis of the articles that were shared by both parties. These articles are summarized by an ellipsis (mean and standard deviation for NewsGuard and the honesty component) that has been added to Figure 3 (Panels C and D). The analysis shows that the articles shared by both parties are of average belief-speaking and truth-seeking, and generally involve high-quality sources.

Clarifications

1. Representatives and Senators often have multiple accounts (direct personal account, campaign account, senatorial account etc) -- how were these labeled. In the case of some representatives, they use their personal as their primary, whereas others use their senatorial/congressional alongside their personal accounts for different purposes. How were these reconciled and if only one account was used per politician, which was chosen and why?

This is a good point. We have clarified how handles were collected.

2. I had some trouble parsing Extended Data Figure 2. First, it appears that the patterns for negative emotion, in particular, are very bursty, with sudden extreme dips followed by highs, including a period in 2014 where negative belief-speaking is extremely high among Democrats. I am not sure I read this right, but if so, what explains these?

In the revision, we have moved those figures to the supplement (Figures S3 and S4). The reviewer is correct in noting the "burstiness" of the pattern, including the high negative belief-speaking peak in 2014 for Democrats, but we are disinclined to attach much credence to that pattern. The accompanying uncertainty band (95% confidence interval) is very broad and the bands of the two parties overlap, so we do not believe much can be made of this early pattern. This is different for the period after 2017, at which point far more data became available and the pattern stabilized.

3. On the inter-rater reliability test, the examples from the annotation guide felt like they were a bit difficult to parse, and it would be helpful to know if there was a debriefing with the raters on what they felt they were getting right or wrong once they reconciled their results with each other. Also, I am assuming that besides the basic classification into the three categories, the raters were judging if the tweet was in fact a negative-emotion belief-speaking.. and so on -- is this correct? Was the Krippendorff's α was computed on a binary agree/disagree (on say belief speaking... etc)? While I find the authors' argument for removing seeking-understanding acceptable, I'm still curious about what the authors feel about the lower values of the inter-rater tests, especially given the nature of sarcasm, innuendo, and indirect speech in political discourse. Also was inter-rater ground truth done following the LIWC positive/negative emotion scoring to confirm automated results?

We have completely reworked our validation methodology, so this comment no longer applies. See our response to Reviewer 2 below for details.

4. Related to the above, do authors have any commentary on the incidence and quality of tweets that did the opposite of truth seeking while using the words (say for instance "proof", or "evidence" used as a discursive means to make belief-speaking seem more evidentiary)

This is a very good point. Unfortunately it is difficult to address because it requires an inference about intentionality: we can identify high truth-seeking tweets that point to low-quality information (although there are, of course, fewer of those than point to high-quality information), but that does not resolve the conundrum of whether those tweets were designed to mimic truth-seeking by their creator or whether this is simply a coincidence that one would expect with noisy statistical data.

It should also be noted that our new method of keyword validation ensures that each keyword is seen to be representative of belief-speaking rather than truth-seeking (or vice versa), so unlike the original selection of keywords, the present set has been judged by random participants to be predominantly of the assigned type. This does not preclude the possibility of using truth-seeking words "by stealth" to camouflage belief-speaking, but at least we can say that it is unlikely to occur by chance alone.

References:

I would encourage authors to look beyond Euro/American research.

Indeed. This is a very important point, and we now point to this issue in our concluding sections. We recently published a paper that looked beyond the U.S. and also considered tweets by British and German parliamentarians (Lasser et al., 2022, PNAS Nexus, DOI: 10.1093/pnasnexus/pgac186) but that was admittedly only a baby step. We now have a section in the Discussion about the need to look beyond the US (and WEIRD countries more generally).

Two things come to mind. First, on the 'speaking their mind' track, one smaller data set approach the authors may find interesting is Gonawela et al (2018) "Speaking their Mind: Populist Style and Antagonistic Messaging in the Tweets of Donald Trump, Narendra Modi, Nigel Farage, and Geert Wilders"

We are grateful for this pointer. We now cite this paper in the revision.

Second, authors can also look at work that has done a two-party political discourse analysis using large scale electoral data, using hashtags in Panda et al Topical Focus of Political Campaigns and its Impact: Findings from Politicians' Hashtag Use during the 2019 Indian Elections

This is also an interesting paper although we found it less pertinent than Gonawela et al.

Reviewer 2

Remarks to the Author:

Below I share my assessment of the study entitled "New conceptions of truth foster misinformation in online public political discourse", which was submitted to Nature Human Behaviour. The study presents a thorough and insightful analysis, applies state-of-the-art methods, examines highly relevant issues, aims to contribute empirical findings to conceptual debates about conceptions of truth, evidence, and knowledge, integrates different disciplinary approaches from the social sciences (psychology, political communication, STS), and seeks to advance ongoing debates within and beyond the scholarly community about how misinformation, political communication, and partisan ideology relate to each other. However, I have a number of criticisms and concerns. Some of these

could be addressed in a substantial revision of the paper, but some weaknesses are, in my view, too severe to be fixed easily. These weaknesses decrease the overall quality of the paper substantially and outweigh the above-mentioned merits, which is why I cannot recommend publication in NHB.

We have completely reworked the analyses which we believe addresses the reviewers (valid) concerns from the first round.

In the following I provide detailed comments. Most of these revolve around the conceptual framing of the study, the interpretation of results, and the conclusion. I am not an expert in (automated) social media analyses, so I'll refrain a bit from commenting on the methodological and analytical procedures.

The conceptualisation of honesty vis-à-vis authenticity, veracity, accuracy etc.

There are a lot of terms and concepts in the paper's frontend which, in my view, are used somewhat interchangeably, such as "truth", "evidence", "accuracy", "honesty", "authenticity", or "veracity". Some of these terms are also (implicitly) opposed to "misinformation", "lying", and "evidence-free". I'd encourage the authors to disentangle these terms, which may mean explaining some of them briefly (particularly "misinformation", which was not defined although it is key to take present study), using them more consistently, or just using fewer different terms.

We agree that our introduction was unfortunately quite loose in terminology. We have reworked the introduction completely and have taken great care to be consistent and clear in our terminology. For example, we now explicitly discuss the distinction between honesty and truth.

Relatedly, I am sorry to say that I was a bit confused by the authors' understanding of (dis)honesty: I can't see how honesty can be valued as "a signal of authenticity" (p. 2), also because the reference [8] does not say anything about (dis)honesty, rather about lying. Isn't it then rather that in some circumstances, people may even come to value *rejection of allegedly elitist evidence/truths* as a signal of authenticity?

Precisely—that is what we were trying to express in that section. We believe that the ambiguity identified

by the reviewer was a direct result of our undisciplined nomenclature, which we have now tightened up.

After all, Cooper et al.'s honesty model – even if useful for the present study – did not really help me much in understanding the concept of honesty nor in disentangling honesty from authenticity or accuracy, for example. Moreover, I am unsure if being honest can be considered an “‘establishment’ norm” (p. 2). Some would not agree with this, because honesty can be conceived as a general virtue and is not precluded to a societal or political establishment.

Again, this is an interesting point that the revision has addressed. In particular, we agree that truth telling is perhaps best understood as an “established norm” (as indeed Hahl et al., 2018, refer to it) rather than an “establishment norm”. However, this does not make a difference to our argument because it remains the case that (established) norm violation is a tool for populist politicians to present themselves as an “authentic champion of the people”.

The theoretical argument on (new) conceptions of truth

The authors argue that a “new ontology of truth” has emerged (p. 2). However, I am not sure if this ontology is actually new: “Alternative” or anti-hegemonic conceptions of truth and evidence – such as those analysed in the present study – have been present ever since (see van Zoonen, 2012, for example).

We agree, of course, that the “new” ontology is not entirely new—after all, we linked the belief-speaking notion to the fascism of the 1930s.

What might be indeed new (at least for some contexts like the US) is an increasing salience of alternative truth conceptions in political discourse;

Precisely, what is new is the relatively brazen attempt by political actors to invoke concepts such as “alternative facts” or to label evidence as “fake news”. We have clarified this in the revision.

not least due to social media, which help spreading and conveying salience to these conceptions, as they allow for circumvention of established gatekeepers, for example (Gerbaudo, 2018; Hopster, 2021). This assumption, however, has not been considered or discussed sufficiently, however. After all, the particularities of social media (and Twitter specifically) are barely discussed throughout the study, albeit the specificities of this communication environment are key to present study.

This is a tricky issue, and we may not be able to fully comply with the reviewer's suggestion. First of all, we agree that social media is critical to the permeation of misinformation (of whatever type). We also agree that social media are a disruptive technology, and the removal of gate keepers has profound implications. Indeed, we have published extensively on the impact of online technologies on cognition and democracy, most recently here: <https://sks.to/techdem>. However, it does not follow that this ought to be a topic of discussion for the present paper. On the contrary, we worry that a discussion of the role of social media would dilute the focus of the paper, which at present is about political communication and how it relates to the quality of information being shared. We therefore continue to side-step the social media issue, except to note that social media offers politicians the opportunity for unfiltered communication with their constituents.

We also believe that our position is supported by the fact that the information being shared by politicians (i.e., the articles being linked to in the media) follows the same regularity as the tweets themselves – that is, compare bottom panels of Figure 3 to the top panels. The parallelism between conventional media (bottom) and social media (top) supports our contention that, in this instance, we can talk about political communication without requiring a detailed analysis of social media.

Relatedly, I think the authors might find a better title for the study, as I find the claim “conceptions [...] foster misinformation” somewhat unintelligible, also because it insinuates a causality that I don't think is supported by the results. Maybe it might be more accurate to use a title like “Misinformation links to alternative truth conceptions in online political discourse” or “Political misinformation and belief-speaking fuel each other” or something like this?

We appreciate the suggestions and we have settled on a new title, which is “From Alternative conceptions of honesty to alternative facts in communications by U.S. politicians”.

Relating truth-seeking to democracy and belief-speaking to fascism

This might be a minor point, but I'd be much more careful with implicitly equating the opposition of belief-speaking and truth-seeking with the opposition of fascism and democracy (see p. 3). Have belief-speaking ontologies really evolved out of fascist ideologies (as it is insinuated in “has roots in..”)? Perhaps belief-speaking has been a feature or concomitant of fascist political thought, but I assume the same is true for some manifestations of nationalism, authoritarianism, or populism.

We are quite confident of the link between variants of contemporary belief speaking and the “intuitive” approaches to truth that characterize fascism (the references cited in connection with this argument confirm the link). But we also agree with the reviewer that similar links exist to other forms of ideology such as populism, which we now also mention, together with one of the papers recommended by the reviewer above.

Introduction of research questions

While some aspects of the research questions (p. 4) have been well introduced and explained, other aspects lack a proper introduction, let alone a rationale for why the authors focused on them. For example, why did they choose to investigate US congress members? Do the US represent a special/characteristic or otherwise comparable/representative case? Why did the authors study partisan differences? Is there any reason to assume that they matter? (Sure there are, but the authors should at least briefly explain them)

We have addressed this in the revision. Our research questions are more tightly specified now.

Embedding in extant research

So far, only very few *empirical* studies on misinformation, political communication, partisanship, and social media have been cited. I encourage the authors to refer to some of these studies at least briefly in order to embed their findings and interpretations in the current literature.

In response, we have added additional citations to the relevant literature where it seemed necessary or useful. In particular, we have spent more time and care interpreting our results (see detailed response below to a related point by the reviewer).

Very low intercoder reliability

The authors find Krippendorff's Alpha values of .42 and .53 (with confidence intervals going as low as .30 and .31, respectively). This is, in my view, a substantial problem. The authors do suggest that Alpha values lower than .667 may be acceptable for ground-breaking research, citing Geiß (2021) and Riffe et al. (1998). But after having a closer look at these texts, I came to assume that

Alpha < .667 is not acceptable even for novel and ground-breaking studies:

Geiß (2021) states:

"Krippendorff (2004a) suggests that Krippendorff's α 's (α_K 's) as high as .800 are necessary for trusting the coding, while α_K 's between .800 and .667 may suffice for drawing preliminary conclusions. Values lower than .667 are characterized as generally unacceptable. Riffe, Lacy, and Fico (1998) agree to the limits proposed by Krippendorff (2004a) but state that in exploratory and ground-breaking research lower values may suffice."

However, Riffe et al. (1998, p. 154) point out:

"However, Krippendorff (2004a) added that variables with Alphas as low as .667 could be acceptable for drawing tentative conclusions. The lower coefficient would be appropriate for research that is breaking new ground with concepts that are rich in analytical value."

These paragraphs might leave some space for speculation, but in my reading, this means that values lower than .667 are barely acceptable, even for novel research. Therefore I have strong concerns about the reliability of the analyses of the present study. When looking at the keyword lists for the three honesty components, my concerns intensified even more: I don't mean to nit-pick on single keywords - but I think actually a lot of them barely tap the honesty component that they mean to capture. For example, "know", "observe", and "think" could well fit truth-seeking component, whereas "claim", "tentative", or "virtually" could also fit the belief-speaking component. Perhaps this is one of the reasons for the very low Krippendorff's Alpha values.

We have completely reworked our approach and this analysis has been removed and replaced by a far more satisfactory approach. Specifically, we now validated the keywords by recruiting a sample of 50 participants who were asked to rate the extent to which each candidate keyword was representative of belief-speaking and truth-seeking. We then compared the ratings on the two dimensions for each word and selected only those that were significantly different for inclusion in the final dictionaries. In consequence, in our new dictionaries, every word has been rated to be significantly higher on the relevant honesty dimension than on its counterpart.

Unequal lengths of keyword lists

I think it is indeed a notable limitation that the keyword lists for belief-speaking and truth-seeking differ. The authors do point out

that this means that the frequency figures of each category may not be comparable. Moreover, however, I think this imbalance introduces further bias into the analyses: For example, all analyses on belief-speaking tweets may be less reliable/robust than the truth-speaking analyses – perhaps they may be considered underpowered in some sense. Also, the frequencies in Figures 1B and 1C are barely comparable because of this imbalance.

Our new dictionaries are of equal length, so this point no longer applies.

Interpretation and discussion of findings

The authors make several interesting and exciting findings – but barely discuss many of them. For example, they find that Republican truth-seeking keywords are linked to controversial matters or economic issues, whereas Democrat truth-seeking texts almost exclusively concern scientific topics (p. 6). The authors do not elaborate on this finding, although I think it is very relevant for the present study: To me it appears as if Republican truth-seeking is related to issues where there are no objective benchmark truths (or no endeavours by scientists/experts/etc to find such truths; e.g., tax, impeachment), whereas Democrat truth-seeking is indeed related to issues where there are consensual truths (e.g., climate).

This pattern has changed slightly owing to the use of a new approach to the analysis (see last point in this letter in reply to Reviewer 3 for details). Nonetheless, it is very interesting and we now discuss it in the revision.

This suggests to me that Republican truth conceptions tend to rely on an understanding of truth that is rather related to practical questions of daily or societal life, whereas Democrat truth conceptions tend to rely on an understanding of truth that is related to epistemological questions.

This is a very interesting point. We agree that our discussion was insufficient, and we have taken great care to improve it in the revision.

Also, I'd encourage the authors to discuss the very interesting finding that approx. the same increase in belief-speaking tweets occurred for both parties, which is somewhat at odds with the popular narrative that only Republicans have come to promote their

beliefs regardless of evidence-based truths in political discourse.

This is an interesting point which we now address. The important insight from this pattern of results is that belief-speaking does not necessitate abandoning accuracy and evidence. It is, however, a gateway to low-quality information that politicians can employ to (likely) justify sharing of low-quality information. Again, we try to tease this apart in more detail in the revision.

Coming to the conclusion section, I think it has substantial need for improvement and additions, for a number of reasons:

- Much of it is a summary of the findings
- It barely connects to the theoretical assumptions outlined in the introduction (e.g., the honesty model)
- It does not pick up the introductory argument on dishonesty (or criticisms of allegedly elitist knowledge claims) being an authenticity marker
 - It does not reflect on the role of social media
 - It does not discuss the specificity of the US context

We thank the reviewer for providing this list of omissions. We have completely reworked the Conclusions to explore the data in a more meaningful way.

All these additions would require more space, of course - which would be available if the authors, for example, report more briefly on the sensitivity analyses (NYT corpus and LIWC text analysis, p. 5-6). These analyses are very worthwhile, but I think they could be moved to the appendix or methods section, and only briefly be summarized in section 2.

We agree, and much of the material that was previously in the paper has been moved to the supplement. We believe that this has served to bring the main story line into sharper focus.

References

- Geiß, S. (2021). Statistical Power in Content Analysis Designs: How Effect Size, Sample Size and Coding Accuracy Jointly Affect Hypothesis Testing - A Monte Carlo Simulation Approach. *Computational Communication Research*, 1(3).
<https://doi.org/10.5117/CCR2021.1.003.GEIß>
- Gerbaudo, P. (2018). Social media and populism: An elective affinity? *Media, Culture & Society*, 40(5), 745-753.
<https://doi.org/10.1177/0163443718772192>

Hopster, J. (2021). Mutual affordances: The dynamics between social media and populism. *Media, Culture & Society*, 43(3), 551-560.

<https://doi.org/10.1177/0163443720957889>

Riffe, D., Fico, F., & Lacy, S. (1998). *Analyzing Media Messages: Using Quantitative Content Analysis in Research* (2nd ed.). Taylor and Francis.

van Zoonen, L. (2012). I-Pistemology: Changing truth claims in popular and political culture. *European Journal of Communication*, 27(1), 56-67. <https://doi.org/10.1177/0267323112438808>

Reviewer 3

Remarks to the Author:

This paper studies an important problem in the quality of political statements using Twitter data.

I have two major concerns:

- 1) The paper categorizes Twitter into belief speaking, truth-seeking, and others. How the categorization relates to misinformation is not clear.

Figure 3 in the revision presents an overview of the relationship between the two types of honesty and the quality of information being shared. We hope this is sufficiently clear now.

The paper uses NewsGuard to measure the correctness. However, the conclusion seems net conclusive based on Figure 3 with quite some differences between the two parties.

We are unsure what the reviewer means by "net conclusive."

Also, the usage of NewsGuard is not perfect as it is also an estimator.

It is impossible to measure information quality without using an estimator. There is evidence that NewsGuard correlates highly with other estimators of information quality; see, e.g. Lasser et al., 2022, PNAS Nexus, DOI: 10.1093/pnasnexus/pgac186. We report a similar validation using an alternative index

in the Supplementary Information accompanying the revision.

- 2) The way to categorize the belief-speaking twitters and truth-seeking twitters is not scientific. The categorization is simply counting the number of word appearances. The results can be largely influenced by the selected keywords. The robustness is not clear which leads to untrustworthy results.

We consider this judgement to be a little harsh, given that keyword-based methods of text analysis continue to enjoy prominence in the literature (see, e.g., Iliev et al., 2016; Bollen et al., 2021). Moreover, it no longer applies to the revision because we have abandoned the word count method and instead adapted a distributed dictionary representation (DDR) approach. On this approach, an embedding is created for each dictionary word using the GloVe algorithm pretrained on 840 billion tokens. The embeddings follow a distributed semantics approach and permit computation of the similarity (i.e., cosine) between the dictionaries and each document under consideration (i.e., a tweet or a New York Times item). In consequence, our analysis now relies on a graded measure of similarity rather than a count-based classification. Our new DDR approach embodies recent approaches to natural language processing.

References

*Iliev, R., Hoover, J., Deghani, M., & Axelrod, R. (2016). Linguistic positivity in historical texts reflects dynamic environmental and psychological factors. *Proceedings of the National Academy of Sciences*, 113(49), E7871-E7879.*

*Bollen, J., Ten Thij, M., Breithaupt, F., Barron, A. T., Rutter, L. A., Lorenzo-Luaces, L., & Scheffer, M. (2021). Historical language records reveal a surge of cognitive distortions in recent decades. *Proceedings of the National Academy of Sciences*, 118(30), e2102061118.*

Decision Letter, first revision:

1st February 2023

Dear Professor Lewandowsky,

Thank you once again for your revised manuscript, entitled "New conceptions of truth foster misinformation in online public political discourse," and for your patience during the re-review process.

Your manuscript has now been evaluated by Reviewers 2 from the original round of review, as well as a new Reviewer (Reviewer 4) with expertise in natural language processing. All reviewer feedback is included at the end of this letter. Although the reviewers found your manuscript to have improved during revision, they also raise some important outstanding concerns. We remain very interested in the possibility of publishing your study in Nature Human Behaviour, but would like to consider your response to these outstanding concerns in the form of a revised manuscript before we make a decision on publication.

1. Reviewer 4 raises important concerns about the validation of your keyword-approach, and suggests that validation should also occur on the document-level. Reviewer 2 raises concerns about whether participants paid attention to the task. Please follow Reviewer 4's advice and collect additional data to validate the reliability of the approach on the document-level as well. When conducting this additional validation survey, please use attention checks to address Reviewer 2's concerns, and describe these in full in your revised manuscript.

2. In addition, Reviewer 4 mentions that they do not follow all details about the regression models. Please carefully revise your manuscript, clearly motivate analytical choices, and if necessary, provide additional analyses. Also make sure that the figures and interpretations of your findings are clearly described.

In sum, we invite you to revise your manuscript taking into account all reviewer and editor comments. We are committed to providing a fair and constructive peer-review process. Do not hesitate to contact us if there are specific requests from the reviewers that you believe are technically impossible or unlikely to yield a meaningful outcome.

We hope to receive your revised manuscript within 4-8 weeks. I would be grateful if you could contact us as soon as possible if you foresee difficulties with meeting this target resubmission date.

- Include a "Response to the editors and reviewers" document detailing, point-by-point, how you addressed each editor and referee comment. If no action was taken to address a point, you must provide a compelling argument. This response will be used by the editors and reviewers to evaluate your revision.
- Highlight all changes made to your manuscript or provide us with a version that tracks changes.

[REDACTED]

We look forward to seeing the revised manuscript and thank you for the opportunity to review your work. Please do not hesitate to contact me if you have any questions or would like to discuss these revisions further.

Sincerely,

Samantha Antusch

Samantha Antusch, PhD
Senior Editor
Nature Human Behaviour

Reviewer expertise:

Reviewer #2: polarization ; communication studies

Reviewer #4: NLP ; (S)BERT

REVIEWER COMMENTS:

Reviewer #2:
Remarks to the Author:

First, I thank the authors for their kind and detailed response to the reviewer comments. Second, I greatly appreciate the efforts the authors took to revise the paper. In my view, they did a very good job; for example, the reworked introduction and conclusion sections are now much more elaborate, appealing, and convincing. Also, I appreciate that the authors followed my suggestion to use more consistent terminology. That being said, I am fine with the decision of not putting too much weight on social media as a political communication environment – I see the arguments provided in the response letter. After all, this decision may ensure that the paper keeps being appealing to a broad audience such as that of NHB.

Most importantly, I am very satisfied with the new analysis that asked 50 participants to rate the extent to which each candidate keyword was representative of belief-speaking and truth-seeking. I do not see many things to criticize here – I just wondered (given that Prolific and similar platforms have received some criticism with regards to satisficing, speeding, cheating etc. of participants) if there were any measures to assure that these 50 individuals were attentive, read the instructions carefully,

and did not speed through the task?

Moreover, I'd encourage the authors to relate their findings at least briefly to recent developments in the U.S. (congress elections and turbulent election of McCarthy as speaker of the House of Representatives) and on Twitter (changes to Twitter as a political communication environment, now after Elon Musk's take-over?). Would the main findings replicate today? Can the trend the study shows be assumed to continue?

In sum, the revisions improved the paper substantially, and I recommend accepting it.

Reviewer #4:

Remarks to the Author:

The manuscript reports on the results from an analysis of two notions of "honesty" as expressed in a complete dataset of tweets by all US Congress members (2011-2022). The authors distinguish between two notions of honesty, "belief-speaking" (BS) and "truth-seeking" (TS). The authors define a heuristic method for scoring a document (tweet or Web page) w.r.t. these two notions of honesty, and they then perform a longitudinal analysis of how their prevalence has changed over time, as well as regression analyses in order to investigate whether each honesty notion is associated with citing more or less trustworthy sources (via URLs embedded in a tweet).

The authors find that both BS and TS have become more prevalent over time, and that, among Republicans, BS is associated with citing less, and TS with citing more, trustworthy sources, whereas among Democrats, there is no significant effect.

I would like to commend the authors for tackling a hard-to-grasp concept such as honesty in a data-driven fashion. This is always a big challenge, and I find the authors' approach intuitively appealing. I also appreciate the multiple sanity checks that the authors conducted. Overall, I found this paper to address an important phenomenon in an interesting fashion.

I don't see fundamental flaws in the work, but certain aspects leave me not 100% convinced, and I couldn't understand all the details, as explained next. Thus, although the work makes very interesting contributions, I'm not championing the paper for being published in its present state.

Although I believe the keyword-based approach may be appropriate, its validation could be more straightforward. The authors asked crowdworkers to rate individual *keywords* w.r.t. to how well they capture BS vs. TS, but what the authors really care about are BS and TS scores at the level of *documents* (tweets or news articles) -- so I think a more appropriate validation should be done at the document level, not at the word level: e.g., ask crowdworkers to rate TS vs. BS for a given document (not word), and define performance as the correlation of these ground-truth scores with the scores obtained for the same document automatically via the keyword-based method. The reason I'm suggesting this is that individual words may be ambiguous, it might depend on the context whether a given word indicates BS or TS (or none), and we don't even care about individual words for the purpose of the present analysis -- what matters is whether we can score entire documents correctly, so evaluating directly at the document level would be the straightest shot in terms of evaluation.

Given that we care about document-level TS and BS scores, one might also consider fitting automated scoring models via supervised machine learning: collect human BS/TS labels via crowdsourcing (as described above), then fit a simple bag-of-words model (e.g., logistic regression, support vector machine, etc.) on a training set, and finally apply the fitted model on the full dataset to automatically score all documents. I imagine that such a model would have better performance than a purely heuristic, keyword-based approach, but I'm aware that machine learning would also lead to further issues (incl. interpretability), so I see this comment rather as a suggestion for the authors than a criticism of their approach.

The authors write that "The analysis of New York Times content thus supported the validity of our dictionaries because truth-seeking and belief-speaking predominated in science and opinion, respectively — exactly as would be expected." But if I understood correctly, the empirical findings are not exactly as expected, since the "politics" category doesn't score on a middle ground between "OpEd" and "science", as the authors had expected (p. 18: "We expected articles in the politics cluster to fall in between"), but rather lowest w.r.t. both BS and TS. Might this be overselling the intuitive appeal of the scoring heuristic? At least, "exactly as would be expected" might not be the appropriate conclusion.

In a similar vein, I found it counterintuitive that "Both belief-speaking and truth-seeking are negatively correlated with the "analytic" [...] language components" -- why is "analytic" negatively correlated with truth-seeking? I'd have expected the opposite, intuitively. This may not necessarily invalidate the keyword-based scoring heuristic, but it is a counterintuitive results that further decreases the conclusion of "exactly as would be expected".

I was also uncertain whether to trust Fig. 2: panels A and B look identical to one another, and so do panels C and D. Is this a bug or a feature? It seemed unlikely to me that Democrats and Republicans would expose *exactly* the same distributions for both BS and TS.

I didn't understand all the details about the regression models. (This might well be due to my limited expertise, which might, however, make me a suitable audience for which sufficiently clear explanations should be provided.) When regressing NewsGuard (NG) scores against BS/TS scores of the tweets themselves, a mixed-effects model was used, whereas when regressing NG against BS/TS scores of the linked news articles, fixed-effects OLS was used. Why different models? Aren't the two settings exactly analogous, the only difference being on what texts BS/TS scores were computed? Why does this call for different regression models (Eq. 1 vs. Eq. 2)? Also, why does Eq. 1 include interactions between BS and TS? And why does Eq. 2 not include those interactions? How do the interaction coefficients (BS*TS) play into the visualization of Fig. 3, which seems to show only the coefficients for BS or TS on their own (and when interacted with the party indicator), but no interaction of BS with TS?

Some aspects of Fig. 1 remained unclear to me. What causes the X-shaped structure of the plot, with plenty of points in the center around the origin, as well as in the corners, but no points in between? For instance, why are there no bipartisan BS/TS words (around coordinate (0,1) or (0,-1))? The description of how Fig. 1 was produced (Sec. 7.5) didn't help me much in understanding it. For instance, I didn't understand how precision and recall come into play ("These values are defined by the package author as precision and recall, respectively"). I also didn't grasp the definition of "SFS" in

Sec. 7.5: in the case distinction (I'm using "x" instead of "SFS^x" here, and analogously for "y"), we choose x if $x > y$, and we choose 1-y if $x < y$. The first condition says "pick the bigger one of x and y", whereas the second condition says "pick one minus the bigger one of x and y", which seem contradictory. If x/y were replaced by Republican/Democrat, it might make sense (not sure), but when introducing x and y as free variables, I don't know how to interpret the definition.

Some more comments:

- It would be interesting to see NG trustworthiness scores plotted over time. If trustworthiness and BS are linked according to the mechanism posited by the authors (where BS is used to give credence to untrustworthy information) and we see more BS over time, one might expect a decrease in trustworthiness scores. Or could it be that the trustworthiness of shared articles remains constant, but the way in which it is mentioned becomes more BS-heavy?
- The authors observe that both BS and TS increase. Does this really justify the claim of "a new understanding of truth and honesty that has *replaced* reliance on evidence with the invocation of subjective belief"? That statement seems to imply an increase in BS and a decrease in TS, whereas in reality both increase.
- The formulation "with authentic but evidence-free belief-speaking becoming more prominent" may also be misleading. Does the BS score really operationalize "evidence-freeness"?
- p. 6: what does "uniquely identifiable" mean?
- What does the subscript "acc" (p. 7) stand for? A priori, I'd read it as "accuracy", but that seems to not be the case here.
- Fig. 2E-F: are the plotted values micro averages over all tweets, or are they macro averages where the same weight is given to each politician?
- In Fig. 3 articles shared by members of both parties were excluded in the regression analysis. Doesn't this bias the findings? Might it be more straightforward to include such articles twice, once per party?
- It would be nice to drill further into the data to understand the causes of the shifts better. For instance, what role do outliers play? That is, what portion of the effects is due to politicians that tweet particularly much? Also, who are the politicians contributing most to the increase in BS and TS, respectively? Similarly, which dictionary words are most responsible for the increase in BS and TS, respectively? This would be useful to know in order to interpret the results.
- Sec. 7.2 briefly mentions fasttext in the beginning, but then doesn't come back to it, and it remains unclear how fasttext is used as part of the methodology.
- The last sentence of Sec. 7.3 mentions that using word2vec or fasttext instead of GloVe yields "similar results", which is vague. It would be good to show those results in the supplemental material.
- The abstract says: "We show that in tweets by conservative members of Congress, an increase in

belief-speaking of 10% is associated with a decrease of 13.7 points of quality [...]", but remains silent about liberals. I'd recommend also stating explicitly that no significant effects were found for liberals.

Author Rebuttal, first revision:

We thank the reviewers for their insightful comments that helped us to improve the methodological rigour of the paper.

We also note that thanks to reviewer #4, we spotted a deficiency in our data collection pipeline. As originally noted in the Method section, we collected *all* tweets for each account—however, because we initially used the default non-academic endpoint of the Twitter API we were only provided with the last 3200 tweets of every account (this is Twitter's interpretation of "all" for that endpoint). This default endpoint was the only one available until late January 2021, at which point an academic endpoint became available that no longer had an upper limit but that required a flag to override the default. When we commenced our analysis in summer 2021, we had not yet updated our pipeline code.

We have now upgraded the analysis by re-collecting all data using academic access to the Twitter API, which now interprets "all" to mean *all* available tweets. We also took that opportunity to extend our corpus to December 31 2022 (it was March 16 2022 at the previous round).

This upgrade in our corpus did not discernibly affect the results although it considerably increased the size of our corpus from the initial 1.8 million to now 3.9 million tweets. We have updated all analyses reported in the article accordingly and report the initial analysis with the restricted data set as robustness analysis in the supplement (Section S8), as suggested by reviewer #4 in point (16a).

In addition to expanding our corpus, we have also improved various aspects of our analysis. First, we noticed that the similarity scores for belief-speaking and truth-seeking depend in part on the length of texts. Given that Twitter increased the maximum length of tweets from 180 to 280 characters in 2017, this implies that a portion of any increase in similarity scores over time might be due to the increase in the length of tweets. To account for this dependency, we now correct similarity scores for the length of the document (i.e., tweet). To achieve this, we first fit two linear models that predict the similarity to belief-speaking and truth-seeking, respectively, based on the length of a document alone. We then subtract these predicted similarity scores from the belief-speaking and truth-seeking similarity scores measured for the given document. These residual similarity scores are length-corrected and hence unaffected by exogenous

variables such as the change in character limit. We use these length-corrected residual scores in all analyses.

Second, we noticed that the GloVe and word2vec embeddings we used to calculate similarity scores included the word “seem” in their list of stopwords, thereby removing it from the original analyses. This is problematic because “seem” appears in our belief-speaking dictionary. To address this issue, we edited the default stopwords list and have created new versions of the GloVe and word2vec embeddings we use in our analyses that do not exclude the word “seem”.

Lastly, we took a closer look at the dictionaries used to measure negative emotions in LIWC22 (i.e., the version of LIWC published in 2022). It turns out that there was a substantial change in the words included in the dictionary between LIWC15 (the original version from 2015) and LIWC22, as the update was redesigned to measure correlates of negative emotions without necessarily measuring negative sentiment. We discovered that the correlation of negative emotion scores on our material between LIWC15 and LIWC22 was surprisingly low, which made us reconsider the appropriateness of LIWC22. We therefore exchanged the LIWC measurement of positive and negative emotions in the previous version of the manuscript for a measurement of positive and negative sentiment using VADER (Hutto et al. 2014, 10.1609/icwsm.v8i1.14550). VADER is a sentiment measurement that was tailored for social media texts and generally outperforms LIWC in sentiment analysis tasks (Hutto et al. 2014). We therefore consider VADER to be more appropriate for the statistical analysis of sentiment. We retain LIWC for measuring the “analytic”, “authentic” and “moral” dimensions, however, because LIWC is, to our knowledge, the only tool available for measuring these speech components.

We note that **all principal patterns in the results and interpretations of our work remain unchanged by these changes in the underlying data and analysis approach**. There are however changes in most figures, mainly because the expansion of the corpus has reduced noise in the data.

In the following, we provide the original reviewer’s comments in *italic green* text and our respective answers in normal font in black. Text additions to the manuscript are indicated as underlined text in blue. We have added numbers to the individual comments to facilitate cross-referencing.

Reviewer #2

*First, I thank the authors for their kind and detailed response to the reviewer comments.
Second, I greatly appreciate the efforts the authors took to revise the paper. In my view, they*

did a very good job; for example, the reworked introduction and conclusion sections are now much more elaborate, appealing, and convincing. Also, I appreciate that the authors followed my suggestion to use more consistent terminology. That being said, I am fine with the decision of not putting too much weight on social media as a political communication environment – I see the arguments provided in the response letter. After all, this decision may ensure that the paper keeps being appealing to a broad audience such as that of NHB. Most importantly, I am very satisfied with the new analysis that asked 50 participants to rate the extent to which each candidate keyword was representative of belief-speaking and truth-seeking.

(1) I do not see many things to criticize here – I just wondered (given that Prolific and similar platforms have received some criticism with regards to satisficing, speeding, cheating etc. of participants) if there were any measures to assure that these 50 individuals were attentive, read the instructions carefully, and did not speed through the task?

As a main addition to the manuscript, we have added a document-level validation (see also reviewer #4 point (1)). To this end, we have recruited another round of respondents on Prolific. This time, we added an attention check to the survey: we asked participants to select “5” for both belief-speaking and truth-seeking in a question halfway through the survey. Only one participant failed this attention check and was subsequently excluded.

(2) Moreover, I'd encourage the authors to relate their findings at least briefly to recent developments in the U.S. (congress elections and turbulent election of McCarthy as speaker of the House of Representatives) and on Twitter (changes to Twitter as a political communication environment, now after Elon Musk's take-over?). Would the main findings replicate today? Can the trend the study shows be assumed to continue?

Our expanded corpus (see General Comments above) now extends to 31 December 2022, which covers the period of the midterm elections and more than two month's of Musks reign (he took over on 27 October). Our expanded corpus does, however, fall short of covering the few days of the McCarthy election (3rd-6th January 2023). Overall, having extended the corpus by 9 months until the end of 2022 we feel that the analysis is quite “contemporary” although there is no guarantee that political rhetoric will remain unchanged in the future. We now make an explicit note of this in the Discussion as follows: “Future research is also needed to examine the temporal stability of the patterns we observed here. Although our analysis extended to the end of 2022, thus covering two months of Twitter activity after it was taken over by Elon Musk, there is no guarantee that the platform will remain stable in the future.”

Likewise, in the same way that sharing of misinformation mushroomed after 2016, the long-term trend towards populism may reverse and the sharing of misinformation may become less frequent in the future. Our analysis is therefore best understood as a historical and contemporary picture of political discourse rather than a pointer to the future."

Reviewer #4

The manuscript reports on the results from an analysis of two notions of "honesty" as expressed in a complete dataset of tweets by all US Congress members (2011-2022). The authors distinguish between two notions of honesty, "belief-speaking" (BS) and "truth-seeking" (TS). The authors define a heuristic method for scoring a document (tweet or Web page) w.r.t. these two notions of honesty, and they then perform a longitudinal analysis of how their prevalence has changed over time, as well as regression analyses in order to investigate whether each honesty notion is associated with citing more or less trustworthy sources (via URLs embedded in a tweet).

The authors find that both BS and TS have become more prevalent over time, and that, among Republicans, BS is associated with citing less, and TS with citing more, trustworthy sources, whereas among Democrats, there is no significant effect.

I would like to commend the authors for tackling a hard-to-grasp concept such as honesty in a data-driven fashion. This is always a big challenge, and I find the authors' approach intuitively appealing. I also appreciate the multiple sanity checks that the authors conducted. Overall, I found this paper to address an important phenomenon in an interesting fashion.

We thank the reviewer for their positive comments and encouraging outlook.

I don't see fundamental flaws in the work, but certain aspects leave me not 100% convinced, and I couldn't understand all the details, as explained next. Thus, although the work makes very interesting contributions, I'm not championing the paper for being published in its present state.

*(1) Although I believe the keyword-based approach may be appropriate, its validation could be more straightforward. The authors asked crowdworkers to rate individual *keywords* w.r.t. to how well they capture BS vs. TS, but what the authors really care about are BS and TS scores at the level of *documents* (tweets or news articles) -- so I think a more appropriate validation should be done at the document level, not at the word level: e.g., ask crowdworkers to rate TS vs. BS for a given document (not word), and define performance as*

the correlation of these ground-truth scores with the scores obtained for the same document automatically via the keyword-based method. The reason I'm suggesting this is that individual words may be ambiguous, it might depend on the context whether a given word indicates BS or TS (or none), and we don't even care about individual words for the purpose of the present analysis -- what matters is whether we can score entire documents correctly, so evaluating directly at the document level would be the straightest shot in terms of evaluation.

Following the reviewer's recommendation, we added a document-level validation of the belief-speaking and truth-seeking concepts. To this end, we sampled a total of 60 tweets from endpoints of the two similarity scales (see below) and asked respondents on Prolific (N=50) to rate the tweets on a five-point likert scale according to their perceived representativeness for belief-speaking and truth-seeking. The tweets were sampled such that 20 tweets were from the top belief-speaking and bottom truth-seeking similarity quartile, 20 from the top truth-seeking and bottom belief-speaking quartile, and 20 that belonged both to the bottom belief-speaking *and* truth-seeking quartiles. For every 20 tweets, 10 were sampled from Republican and 10 from Democrat accounts. The instructions given to participants were exactly the same as for the word-level validation, with the only difference being that we replaced the term "word" with the term "tweet". We then created a ground-truth data set from the human ratings by assigning the label "belief-speaking" to every tweet where a majority of human raters rated the tweet as "4" or "5" on the five-point likert scale of representativeness for belief-speaking and all other tweets as "not belief-speaking". We followed the same approach to create a ground-truth data set for truth-seeking.

To assess the performance of our similarity-based belief-speaking "classifier", we created an ROC curve by varying the threshold for belief-speaking similarity to categorise a tweet as "belief-speaking" (akin to varying response criteria in a behavioral study). The area under the ROC curve (AUC) was 0.824. We followed the same procedure for truth-seeking similarity and found AUC=0.772. Given this satisfactory agreement between human raters and our similarity-based classification, we are confident that our measurement instrument generalises well and is able to pick up the relevant concepts in text. We report this additional validation in section 7.4 of the Methods, with additional detail (including the ROC curves and distributions of all human ratings for all tweets) reported in section S3 of the online supplement.

(2) Given that we care about document-level TS and BS scores, one might also consider fitting automated scoring models via supervised machine learning: collect human BS/TS labels via crowdsourcing (as described above), then fit a simple bag-of-words model (e.g., logistic regression, support vector machine, etc.) on a training set, and finally apply the fitted model on the full dataset to automatically score all documents. I

imagine that such a model would have better performance than a purely heuristic, keyword-based approach, but I'm aware that machine learning would also lead to further issues (incl. interpretability), so I see this comment rather as a suggestion for the authors than a criticism of their approach.

We agree that training a supervised machine learning model is a promising alternative approach. We do intend to follow this approach, specifically by fine-tuning a state-of-the-art transformer model on a data set of crowd-sourced belief-speaking and truth-seeking similarity scores for documents, for a follow-up study. Nonetheless, for the current study we keep the existing approach because it showed satisfactory ability to distinguish belief-speaking and truth-seeking and, as the reviewer mentions, interpretability of the model is an asset that we wanted to keep.

(3) The authors write that "The analysis of New York Times content thus supported the validity of our dictionaries because truth-seeking and belief-speaking predominated in science and opinion, respectively — exactly as would be expected." But if I understood correctly, the empirical findings are not exactly as expected, since the "politics" category doesn't score on a middle ground between "OpEd" and "science", as the authors had expected (p. 18: "We expected articles in the politics cluster to fall in between"), but rather lowest w.r.t. both BS and TS. Might this be overselling the intuitive appeal of the scoring heuristic? At least, "exactly as would be expected" might not be the appropriate conclusion.

We agree with the reviewer and rephrased the statement to be more nuanced and also mention the part of our expectations that was not confirmed. The relevant paragraph in the manuscript now reads: The analysis of New York Times content confirmed our expectation of articles in the science category being most similar to truth-seeking while articles in the opinion category being most similar to belief-speaking. It did not confirm our expectation of politics being more similar to truth-seeking than opinion articles and more similar to belief-speaking than science articles.

(4) In a similar vein, I found it counterintuitive that "Both belief-speaking and truth-seeking are negatively correlated with the "analytic" [...] language components" -- why is "analytic" negatively correlated with truth-seeking? I'd have expected the opposite, intuitively. This may not necessarily invalidate the keyword-based scoring heuristic, but it is a counterintuitive results that further decreases the conclusion of "exactly as would be expected".

First of all we need to stress that we had no prior expectations regarding the direction of the correlations of the LIWC measured language components with belief-speaking and truth-

seeking. The statement “*The analysis of New York Times content thus supported the validity of our dictionaries because truth-seeking and belief-speaking predominated in science and opinion, respectively – exactly as would be expected.*” is purely related to the analysis of the New York Times corpus, not to the LIWC analysis. Our only expectation was that the magnitude of the correlations would be small, showing that we are identifying a largely independent construct.

We do, however, agree that the negative correlation of truth-seeking with “analytic” is somewhat counter-intuitive, as is the positive correlation with “authentic”. Nonetheless, the fact that the correlation of belief-speaking with “analytic” ($r = -0.27$) is about double in absolute magnitude than the correlation between truth-seeking and “analytic” ($r = -0.16$) does fit our intuition again. In summary, all of these correlations are so small that we don’t believe they contribute a lot to an interpretation of the new belief-speaking and truth-seeking language components.

*(5) I was also uncertain whether to trust Fig. 2: panels A and B look identical to one another, and so do panels C and D. Is this a bug or a feature? It seemed unlikely to me that Democrats and Republicans would expose *exactly* the same distributions for both BS and TS.*

We double-checked the plot and indeed found that we failed to stratify by party before plotting the distributions. We thank the reviewer for spotting this mistake! We corrected the plot and the distributions are now visually different (as expected) but still qualitatively very similar. We apologize that we did not catch this error before submission.

(6) I didn’t understand all the details about the regression models. (This might well be due to my limited expertise, which might, however, make me a suitable audience for which sufficiently clear explanations should be provided.) When regressing NewsGuard (NG) scores against BS/TS scores of the tweets themselves, a mixed-effects model was used, whereas when regressing NG against BS/TS scores of the linked news articles, fixed-effects OLS was used. Why different models? Aren’t the two settings exactly analogous, the only difference being on what texts BS/TS scores were computed? Why does this call for different regression models (Eq. 1 vs. Eq. 2)?

We use two different models because the settings are not exactly analogous: in the setting where BS/TS scores are computed on the tweet level, there is a clear nesting of tweets within accounts (every tweet is associated with exactly one account but one account can have more than one tweet). Therefore it makes sense to include accounts as groups and use a linear mixed effects modelling approach allowing for random slopes and intercepts for the effect of belief-speaking and truth-seeking similarity on NewsGuard score for each account. (See Eq

1).

In the setting where BS/TS scores are computed on the article level, there is no nesting of articles within accounts, as one article can be linked to in tweets from multiple different accounts. Therefore it does not make sense to include accounts as groups and an ordinary regression model is sufficient. We clarified this choice of models by adding the following sentence in Section “Regression” in the Methods: “Note that we do not fit a linear mixed effects model for the statistical analysis of the articles, since there is no clear nesting of articles within individual Twitter accounts, as a single article can be linked to from multiple accounts.”

(7) Also, why does Eq. 1 include interactions between BS and TS? And why does Eq. 2 not include those interactions?

We actually did include an interaction between BS and TS for the article model reported in Eq. (2) (as also visible in Extended Data Table 3 where we report the estimates).

Unfortunately we mangled the model design formula in Eq. (2) (we also omitted the intercept and the party x BS x TS interactions). We have fixed Eq. (2) to correspond to the model we actually fitted – which is analogous to Eq. (1), except for the within-account nesting – and we thank the reviewer for spotting this mistake in our equation! This was a typographical error that did not affect the analysis.

*How do the interaction coefficients (BS*TS) play into the visualization of Fig. 3, which seems to show only the coefficients for BS or TS on their own (and when interacted with the party indicator), but no interaction of BS with TS?*

Indeed, the interaction coefficients BS*TS do not play into the visualisation in Fig. 3, as we only show the predictions based on the estimates for BS and TS there. This is a deliberate choice to not overload Fig. 3 with information. We visualise the effect of the three-way interaction P*BS*TS in Extended Data Figure 1 to supplement what is shown in Fig. 3.

(8) Some aspects of Fig. 1 remained unclear to me. What causes the X-shaped structure of the plot, with plenty of points in the center around the origin, as well as in the corners, but no points in between? For instance, why are there no bipartisan BS/TS words (around coordinate (0,1) or (0,-1))? The description of how Fig. 1 was produced (Sec. 7.5) didn't help me much in understanding it. For instance, I didn't understand how precision and recall come into play (“These values are defined by the package author as precision and recall, respectively”). I also didn't grasp the definition of “SFS” in Sec. 7.5: in the case distinction (I'm using “x” instead of “SFS^x” here, and analogously for “y”), we choose x if x>y, and we choose 1-y if x<y. The first condition says “pick the bigger one of x and y”, whereas the second condition says “pick one minus the bigger one of x and y”, which seem contradictory. If x/y were replaced by

Republican/Democrat, it might make sense (not sure), but when introducing x and y as free variables, I don't know how to interpret the definition.

We have edited the description of the scattertext plot in the methods for clarity and provided more detail, following the description of the algorithm's author (see <https://github.com/JasonKessler/scattertext#understanding-scaled-f-score>).

The aim of the described formula is to map two scores (one for category x and one for category y) that are both defined in the range [0, 1] to a single score in the range [-1, 1]. To this end, the formula maps the score for category y to [-1, 0], picks the score with the larger magnitude and then rescales it to the new range. The definition does indeed only make sense if x and y are categories, not arbitrary free variables. We agree that the notation with “x” and “y” as superscripts is misleading in that regard, but we chose to keep it since it reflects the original notation of the author in the description of his algorithm.

(9) It would be interesting to see NG trustworthiness scores plotted over time. If trustworthiness and BS are linked according to the mechanism posited by the authors (where BS is used to give credence to untrustworthy information) and we see more BS over time, one might expect a decrease in trustworthiness scores. Or could it be that the trustworthiness of shared articles remains constant, but the way in which it is mentioned becomes more BS-heavy?

We refer the reviewer to Fig. 1 D in our recent and related publication in PNAS nexus (<https://doi.org/10.1093/pnasnexus/pgac186>). Indeed, trustworthiness scores for Republicans have sharply decreased in the last years, analogous to the increase in belief- speaking.

*(10) The authors observe that both BS and TS increase. Does this really justify the claim of "a new understanding of truth and honesty that has *replaced* reliance on evidence with the invocation of subjective belief"? That statement seems to imply an increase in BS and a decrease in TS, whereas in reality both increase.*

This is an interesting point and shows that our phraseology did not convey the intended meaning. We did not mean to imply that BS has replaced TS, but that the spread of misinformation is facilitated by BS in replacement of TS. The full sentence was: “The results support the hypothesis that the current dissemination of misinformation in political discourse is in part driven by a new understanding of truth and honesty that has replaced reliance on evidence with the invocation of subjective belief.” We have rephrased this as: “The results support the hypothesis that the current dissemination of misinformation in political discourse is in part driven by an alternative understanding of truth and honesty that emphasizes invocation of subjective belief at the expense of reliance on evidence”. We hope that this is clearer and no

longer conveys a claim that BS has replaced TS (given that both increase, as the reviewer correctly notes).

(11) *The formulation "with authentic but evidence-free belief-speaking becoming more prominent" may also be misleading. Does the BS score really operationalize "evidence-freeness"?*

This is a good point. We have rewritten this as “...we show that politicians’ conception of truth has undergone a distinct shift, with authentic belief-speaking that may be decoupled from evidence becoming more prominent and more differentiated from explicitly evidence-based truth seeking.” We hope that this captures the spirit of our findings more accurately.

(12) *p. 6: what does "uniquely identifiable" mean?*

We deleted that part of the sentence since it is redundant with the second part “... do not overlap greatly with existing related measures”.

(13) *What does the subscript "acc" (p. 7) stand for? A priori, I'd read it as "accuracy", but that seems to not be the case here.*

The subscript “acc” denotes an account average. We have added the sentence “Note that \$\langle \rangle_{acc}\$ denotes an account-average.” when we first mention account averages in the manuscript.

(14) *Fig. 2E-F: are the plotted values micro averages over all tweets, or are they macro averages where the same weight is given to each politician?*

Fig. 2E-F show micro averages over all tweets. We have clarified this in the figure caption.

(15) *In Fig. 3 articles shared by members of both parties were excluded in the regression analysis. Doesn't this bias the findings? Might it be more straightforward to include such articles twice, once per party?*

We considered this possibility as well before making the decision of excluding the articles that were shared by both parties. Since only a very small minority of articles – 0.91% (2,462 articles) – is linked to from accounts belonging to both parties, the impact of these articles on the analysis is negligible. We report this in the subsection “News article collection” in the methods, but agree that this is hard to miss. That’s why we included the grey ellipsis in Fig 3. C and D, indicating where the excluded articles fall on the BS – NG and TS – NG planes.

(16) *It would be nice to drill further into the data to understand the causes of the shifts better. For instance, what role do outliers play?*

(a) That is, what portion of the effects is due to politicians that tweet particularly much?

We thank the reviewer for this question which prompted us to calculate more descriptive statistics of our data set. In the process, as noted at the outset of this document, we discovered that we had not, as initially thought, collected the full timeline of each politician's account but rather only the last 3,200 tweets, since we were initially not using the academic endpoint of the twitter API. The default non-academic endpoint of the Twitter API returns the last 3,200 tweets in response to a request for "all" tweets and that endpoint was the only one available until January 2021, a few months before we commenced our analysis.

We re-collected all data using the academic endpoint, increasing the size of the corpus from 1.8 to 3.9 million tweets. We now use the full data set and have updated all our analyses accordingly. We nevertheless kept the initial analysis using only the last 3,200 tweets and now report it in the supplement (Section S11) as a robustness check. This analysis is worth retaining because it has two advantages: first, it de-emphasizes tweets that are further in the past which is desirable because populist discourse surged only in 2016. Second, the limit of 3,200 tweets eliminates the influence of "super-posters" (i.e., politicians who post massively more than others and who might thereby shape the analysis more than their number might warrant.

In the full data set we indeed see that some accounts contribute many more tweets than others – the most prolific account posted 52,055 tweets in the observed time span whereas some accounts only posted a single tweet. The median number of tweets per account is 2,876. The results using the full data set are almost identical to the results using the last 3200 tweets, as can be seen by comparing Extended Data Table 2 to Table S9 in the supplement.

This answers the point (16 (a)) raised by the reviewer by showing that our analysis is robust to capping overly prolific accounts at 3,200 tweets – which is also close to the observed median number of tweets in the full data set. In addition to this robustness analysis, we have added additional descriptive statistics about the number of tweets per account in the supplement Section S11.

(b) Also, who are the politicians contributing most to the increase in BS and TS, respectively?

We added a new section (S12) in the online supplement where we provide descriptive details about the increase in belief-speaking and truth-seeking similarity for individual politicians. We provide two tables listing the top 10 Democratic and top 10 Republican accounts with the highest increase in belief-speaking (Table S10) as well as truth-seeking (Table S11).

(c) Similarly, which dictionary words are most responsible for the increase in BS and TS, respectively?

To answer this question, we created embeddings of every individual keyword in the dictionary using GloVe and then calculated the similarity between embedded individual keywords and embedded tweets. We show the similarity time-series for all keywords in a new section (S14) in the supplement. We see that all keywords in the dictionary show an increase in the observed time period. Nevertheless, there seem to be some keywords like “surely”, “seem”, “feel”, and “obviously” that contribute more to the increase in belief-speaking, while other keywords like “suggest”, “judge”, “signal” and “view” contribute less. For truth-seeking, we see that the words “fact”, “actually”, “prove” and “truth” show a comparatively higher increase in similarity scores than for example the words “trial”, “correction”, “quiz” and “revise”.

(17) Sec. 7.2 briefly mentions fasttext in the beginning, but then doesn't come back to it, and it remains unclear how fasttext is used as part of the methodology.

We use fasttext to compute the cosine similarity between seed keywords and other words in the English language. We then include words with a similarity > 0.75 in the dictionary. We have added the following description at the appropriate place in the methods: “Using the fasttext embeddings, we expanded the seed words to include words that have a cosine similarity score above 0.75.”

(18) The last sentence of Sec. 7.3 mentions that using word2vec or fasttext instead of GloVe yields “similar results”, which is vague. It would be good to show those results in the supplemental material.

We have added a new section (S13) to the supplement, showing the full results of fitting the linear mixed effects model using word2vec and fasttext embeddings instead of GloVe (Tables S12 and S13). We now reference this section at the appropriate point in the methods.

(19) The abstract says: “We show that in tweets by conservative members of Congress, an increase in belief-speaking of 10% is associated with a decrease of 13.7 points of quality [...]”, but remains silent about liberals. I'd recommend also stating explicitly that no significant effects were found for liberals.

As suggested, we added the information to the abstract: We show that for Republicans – but not Liberals – an increase of belief-speaking of 10% is associated with a decrease of 12.8 points of quality (using the NewsGuard scoring system) in the sources shared in a tweet.

Decision Letter, second revision:

11th July 2023

Dear Dr. Lewandowsky,

Thank you for your patience as we've prepared the guidelines for final submission of your Nature Human Behaviour manuscript, "From Alternative conceptions of honesty to alternative facts in communications by U.S. politicians" (NATHUMBEHAV-22071869B). Please carefully follow the step-by-step instructions provided in the attached file, and add a response in each row of the table to indicate the changes that you have made. Please also address the additional marked-up edits we have proposed within the reporting summary. Ensuring that each point is addressed will help to ensure that your revised manuscript can be swiftly handed over to our production team.

We would hope to receive your revised paper, with all of the requested files and forms within two-three weeks. Please get in contact with us if you anticipate delays.

Nature Human Behaviour offers a Transparent Peer Review option for new original research manuscripts submitted after December 1st, 2019. As part of this initiative, we encourage our authors to support increased transparency into the peer review process by agreeing to have the reviewer comments, author rebuttal letters, and editorial decision letters published as a Supplementary item. When you submit your final files please clearly state in your cover letter whether or not you would like to participate in this initiative. Please note that failure to state your preference will result in delays in accepting your manuscript for publication.

In recognition of the time and expertise our reviewers provide to Nature Human Behaviour's editorial process, we would like to formally acknowledge their contribution to the external peer review of your manuscript entitled "From Alternative conceptions of honesty to alternative facts in communications by U.S. politicians". For those reviewers who give their assent, we will be publishing their names alongside the published article.

Cover suggestions

As you prepare your final files we encourage you to consider whether you have any images or illustrations that may be appropriate for use on the cover of Nature Human Behaviour.

ORCID

Non-corresponding authors do not have to link their ORCIDs but are encouraged to do so. Please note that it will not be possible to add/modify ORCIDs at proof. Thus, please let your co-authors know that if they wish to have their ORCID added to the paper they must follow the procedure described in the following link prior to acceptance:

Nature Human Behaviour has now transitioned to a unified Rights Collection system which will allow our Author Services team to quickly and easily collect the rights and permissions required to publish your work. Approximately 10 days after your paper is formally accepted, you will receive an email in providing you with a link to complete the grant of rights. If your paper is eligible for Open Access, our Author Services team will also be in touch regarding any additional information that may be required to arrange payment for your article.

Please note that *Nature Human Behaviour* is a Transformative Journal (TJ). Authors may publish their research with us through the traditional subscription access route or make their paper immediately open access through payment of an article-processing charge (APC). Authors will not be required to make a final decision about access to their article until it has been accepted. Find out more about

Transformative Journals

[REDACTED]

Best regards,
Alex McKay
Editorial Assistant
Nature Human Behaviour

On behalf of

Samantha Antusch

Samantha Antusch, PhD
Senior Editor
Nature Human Behaviour

Reviewer #2:
Remarks to the Author:

Below I will briefly share my assessment of the revised version of the manuscript "From Alternative conceptions of honesty to alternative facts in communications by U.S. politicians".

I thank the authors for revising the manuscript (and the analyses!) once more and explaining the changes in the response letter. I am also grateful to R4 that they have pointed the authors to a flaw in the previous analysis. Regarding my comments on the revised version of the manuscript: I feel that they have been adequately addressed – I particularly appreciate that the authors collected another round of data using an attention check.

My impression of the revised version of the manuscript was already positive, and it has now even improved. Hence, I recommend accepting it.

Reviewer #5:

Remarks to the Author:

Before proceeding, I would like to provide a disclaimer that I joined the review process at a later stage and did not review the paper in its initial round. Therefore, my evaluation is based on how the paper addressed the review comments from the previous round. Please find my assessment below.

This paper presents an interesting and highly relevant contribution that has the potential to expand our existing knowledge in this domain.

The authors curated two dictionaries to differentiate between a truth-seeking, evidence-based concept of honesty and a belief-speaking approach based on intuition, subjective impressions, and feelings. They established the validity of these dictionaries through ratings obtained from human participants. Notably, the prevalence of belief-speaking was observed in opinion pieces in the New York Times, while truth-seeking language was more commonly found in the science section.

The study by the authors offers valuable insights into the prevalence of belief-speaking (BS) and truth-seeking (TS) language over time. They observed an increase in the usage of both language types. Particularly intriguing is their finding that among Republicans, belief-speaking is associated with citing fewer trustworthy sources, whereas truth-seeking is linked to citing more trustworthy sources. This suggests a potential disparity in information quality and source reliability based on the type of language employed.

I would like to commend the authors for their meticulous and evidence-based approach in addressing the complex concept of honesty. Their commitment to rigorous analysis and reliance on data sets a commendable standard for studying such a multifaceted topic. Understanding and quantifying the concept of honesty can be challenging, and the authors have tackled this task with clarity and precision.

The methodological choices they made, along with the robustness checks they conducted, significantly enhance the reliability of their findings.

In conclusion, this paper makes an important contribution by shedding light on the dynamics of language use in political discourse, specifically regarding the distinction between belief-speaking and truth-seeking. The authors' approach is insightful and compelling, revealing the intricate relationship between language choice and the quality of information shared.

In my assessment, this research introduces a novel approach to understand misinformation by leveraging the frameworks of belief-speaking and truth-seeking. Moreover, the authors have effectively addressed all the previous review questions, indicating that the study is well-prepared for publication.

Final Decision Letter:

Dear Professor Lewandowsky,

We are pleased to inform you that your Article "From Alternative conceptions of honesty to alternative facts in communications by U.S. politicians", has now been accepted for publication in *Nature Human Behaviour*.

Please note that *Nature Human Behaviour* is a Transformative Journal (TJ). Authors whose manuscript was submitted on or after January 1st, 2021, may publish their research with us through the traditional subscription access route or make their paper immediately open access through payment of an article-processing charge (APC). Authors will not be required to make a final decision about access to their article until it has been accepted. IMPORTANT NOTE: Articles submitted before January 1st, 2021, are not eligible for Open Access publication. Find out more about Transformative Journals

Once your manuscript is typeset and you have completed the appropriate grant of rights, you will receive a link to your electronic proof via email with a request to make any corrections within 48 hours. If, when you receive your proof, you cannot meet this deadline, please inform us at

rjsproduction@springernature.com immediately. Once your paper has been scheduled for online publication, the Nature press office will be in touch to confirm the details.

With best regards,

Samantha Antusch

Samantha Antusch, PhD
Senior Editor
Nature Human Behaviour